# The decomposition of the higher-order homology embedding constructed from the $k$-Laplacian

**Yu-Chia Chen**
Electrical & Computer Engineering
University of Washington
Seattle, WA 98195
yuchaz@uw.edu

**Marina Meilă**
Department of Statistics
University of Washington
Seattle, WA 98195
mmp2@uw.edu

## Abstract

The null space of the $k$-th order Laplacian $\mathcal{L}_k$, known as the *k-th homology vector space*, encodes the non-trivial topology of a manifold or a network. Understanding the structure of the homology embedding can thus disclose geometric or topological information from the data. The study of the null space embedding of the graph Laplacian $\mathcal{L}_0$ has spurred new research and applications, such as spectral clustering algorithms with theoretical guarantees and estimators of the Stochastic Block Model. In this work, we investigate the geometry of the $k$-th homology embedding and focus on cases reminiscent of spectral clustering. Namely, we analyze the *connected sum* of manifolds as a perturbation of the direct sum of their homology embeddings. We propose an algorithm to factorize the homology embedding into subspaces corresponding to a manifold's simplest topological components. The proposed framework is applied to the *shortest homologous loop detection* problem, a problem known to be NP-hard in general. Our spectral loop detection algorithm complements existing methods of topological data analysis. It scales better than existing methods with no assumptions on the structure of data and is effective on diverse data such as point clouds and images.

## 1 Motivation

The $k$-th *homology vector space* $\mathcal{H}_k$ provides rich geometric information on manifolds/networks. For instance, the zeroth, the first, and the second homology vector spaces identify the connected components, the loops, and the cavities in the manifold, respectively. Topological Data Analysis (TDA) [24, 58] (as well as other early works in this field) has found wide use in analyzing biological [31, 47], human behavior [1, 64], or other complex systems [58]. Even though they easily generalize to $k \geq 1$, additional efforts are needed to extract topological features (e.g., instances of loops) besides ranks due to the combinatorial complexity of the structures that support them.

Spectral methods based on $k$-Laplacians ($\mathcal{L}_k$), by contrast, investigate $\mathcal{H}_k$ in a linear algebraic manner; abundant geometric information can be extracted from the *homology embedding* $\mathbf{Y}$ (the null space eigenvectors of $\mathcal{L}_k$) of $\mathcal{H}_k$. Analysis of the eigenfunctions (of $\mathcal{H}_0$) [14, 37, 40, 49] of the graph Laplacian $\mathcal{L}_0$ is pivotal in providing guarantees for spectral clustering and community detection algorithms in well-separated datasets. Recent advances in this field [4, 12, 48] extend the existing spectral algorithms based on $\mathcal{L}_0$ to $k \geq 1$; however, the theoretical analysis of $\mathbf{Y}$ of $\mathcal{H}_k$, unlike spectral clustering, is less developed, in spite of intriguing empirical results by [21]. Here, we put these observations on a formal footing based on the concepts of *connected sum* and *prime decomposition* of manifolds (Section 2 and 3). We examine these operations through the lens of the (subspace) perturbation to the homology embedding $\mathbf{Y}$ of the discrete $k$-Laplacian $\mathcal{L}_k$ (an estimator of the continuous $k$-Hodge Laplacian operator $\Delta_k$) on finite samples (Section 4). This framework

35th Conference on Neural Information Processing Systems (NeurIPS 2021).

finds applications in, i.e., identifying the *shortest homologous loops* (Section 5). Lastly, we support our theoretical claims with numerous empirical results from point clouds and images.

## 2   Background in Hodge theory and topology

**Simplicial and cubical complex.**   An *abstract complex* is a natural extension of a graph designed to capture higher-order relationships between its vertices. A *simplicial $k$-complex* (used when the data are point clouds or networks) is a tuple $\mathrm{SC}_k = (\Sigma_0, \cdots, \Sigma_k)$, with $\Sigma_\ell$ being a set of $\ell$ dimensional *simplices*, such that every *face* of a simplex $\sigma \in \Sigma_\ell$ is in $\Sigma_{\ell-1}$ for $\ell \leq k$. As a side note, a graph $G = (V, E)$ is an $\mathrm{SC}_1$; and $\mathrm{SC}_2 = (V, E, T)$, which is commonly used in edge flow learning [12, 48], is obtained by adding a set of 3-cliques (triangles) $T$ of $G$. This procedure extends to defining $\Sigma_\ell$ as the set of all $\ell$-cliques of $G$, with the resulting complex called a *clique complex* of the graph $G$. This complex is also known as a *Vietoris-Rips (VR) complex* if $G$ is the $\epsilon$-radius neighborhood graph used in the manifold learning literature [12, 14, 53], The *cubical $k$-complex* $\mathrm{CB}_k = (K_0, \cdots, K_k)$ is a complex widely used with image data. The difference between this complex and the $\mathrm{SC}_k$ is that a $\mathrm{CB}_k$ is a collection of sets of $\ell$-cubes, for $\ell < k$. Note that we write $\Sigma_0 = K_0 = V$ the vertex set and $\Sigma_1 = K_1 = E$ the edge set. $\Sigma_2 = T$ and $K_2 = R$ are the triangle and rectangle set, respectively. Additionally, we define $n_\ell = |\Sigma_\ell|$ (or $= |K_\ell|$) to be the cardinality of the $\ell$-dimensional cells and let $n = n_0$ for simplicity. For more information about building various complexes on different datasets please refer to Otter et al. [43].

**$k$-cochain.**   By choosing an orientation for every $k$-simplex $\sigma_{k,i} \in \Sigma_k$ (or $K_k$), one can define a finite-dimensional vector space $\mathcal{C}_k$ ($k$-*cochain space*[1]). An element $\boldsymbol{\omega}_k = \sum_i \omega_k(\sigma_{k,i})\sigma_{k,i} \in \mathcal{C}_k$ is called a $k$-*cochain*; one can further express $\boldsymbol{\omega}_k$ as $\boldsymbol{\omega}_k = (\omega_{k,1}, \cdots, \omega_{k,n_k})^\top \in \mathbb{R}^{n_k}$ by identifying each $\sigma_{k,i}$ with the standard basis vector $\mathbf{e}_i \in \mathbb{R}^{n_k}$. Functions on nodes and edge flows, for example, are elements of $\mathcal{C}_0$ and $\mathcal{C}_1$, respectively.

**Boundary matrix.**   The $k$-th *boundary matrix* $\mathbf{B}_k$ [35] maps a $k$-cochain of $k$-cells (simplices/cubes) $\sigma_k$ to the $(k-1)$-cochain of its faces, i.e., $\mathbf{B}_k : \mathcal{C}_k \to \mathcal{C}_{k-1}$. $\mathbf{B}_k \in \{0, \pm1\}^{n_{k-1} \times n_k}$ is a sparse binary matrix, with the sign of the non-zero entries $\sigma_{k-1}, \sigma_k$ given by the orientation of $\sigma_k$ w.r.t. its face $\sigma_{k-1}$. Hence, different SC or CB will induce different $\mathbf{B}_k$. For $k = 1$ on either the SC or CB, the boundary map is the *graph incidence matrix*, i.e., $(\mathbf{B}_1)_{[x],[x,y]} = 1$, $(\mathbf{B}_1)_{[y],[x,y]} = -1$, and zero otherwise; for $k = 2$, each column of $\mathbf{B}_2$ contains the orientation of a triangle/rectangle w.r.t. its edges. Specifically, for an SC, $(\mathbf{B}_2)_{[x,y],[x,y,z]} = (\mathbf{B}_2)_{[y,z],[x,y,z]} = 1$, $(\mathbf{B}_2)_{[x,z],[x,y,z]} = -1$, and 0 otherwise; for a CB, $(\mathbf{B}_2)_{[x,y],[x,y,z,w]} = (\mathbf{B}_2)_{[y,z],[x,y,z,w]} = (\mathbf{B}_2)_{[z,w],[x,y,z,w]} = 1$, $(\mathbf{B}_2)_{[x,w],[x,y,z,w]} = -1$, and 0 otherwise. Simplex $\sigma_{k+1}$ is a *coface* of $\sigma_k$ iff $\sigma_k$ is a face of $\sigma_{k+1}$; let $\mathrm{coface}(\sigma_k)$ be the set of all cofaces of $\sigma_k$. The $(k-1)$-th *coboundary matrix* $\mathbf{B}_k^\top$ (adjoint of $\mathbf{B}_k$) maps $\sigma_{k-1}$, as a $(k-1)$-cochain, to the $k$-cochain of $\mathrm{coface}(\sigma_{k-1})$.

**$k$-Hodge Laplacian.**   Let $\mathbf{W}_\ell$ be a diagonal non-negative *weight matrix* of dimension $n_\ell$, with $[\mathbf{W}_\ell]_{\sigma,\sigma}$ representing the weight of the $\ell$-simplex/cube $\sigma$ and $\mathbf{w}_\ell \leftarrow \mathrm{diag}(\mathbf{W}_\ell)$. The weighted $k$-Hodge Laplacian[2] [26] is defined as

$$\boldsymbol{\mathcal{L}}_k = \mathbf{A}_k^\top \mathbf{A}_k + \mathbf{A}_{k+1} \mathbf{A}_{k+1}^\top, \text{ where } \mathbf{A}_\ell = \mathbf{W}_{\ell-1}^{-1/2} \mathbf{B}_\ell \mathbf{W}_\ell^{1/2} \text{ for } \ell = k, k+1. \tag{1}$$

The weights capture combinatorial or geometric information and must satisfy the consistency relation $\mathbf{w}_\ell(\sigma_\ell) = \sum_{\sigma_{\ell+1} \in \mathrm{coface}(\sigma_\ell)} \mathbf{w}_{\ell+1}(\sigma_{\ell+1})$ (in matrix form: $\mathbf{w}_\ell = |\mathbf{B}_{\ell+1}|\mathbf{w}_{\ell+1})$ for $\ell = k, k-1$. Hence $\mathbf{A}_k$ can be seen as normalized boundary matrix. To determine the weight for the $(k+1)$-simplices, one can selected $\mathbf{w}_{k+1}$ to be constant [48] or based on (a product of) pairwise distance kernel (for $k = 0, 1$) so that the large sample limit exists [12, 14]. The first and second terms of (1) are called respectively the *down* ($\boldsymbol{\mathcal{L}}_k^{\mathrm{down}} = \mathbf{A}_k^\top \mathbf{A}_k$) and *up* ($\boldsymbol{\mathcal{L}}_k^{\mathrm{up}} = \mathbf{A}_{k+1} \mathbf{A}_{k+1}^\top$) Laplacians. For $k = 0$, the down component disappears and the resulting $\boldsymbol{\mathcal{L}}_0$ is the *symmetric normalized* graph Laplacian used in spectral clustering [55] and Laplacian Eigenmap [5].

---

[1]We use *chain* and *cochain* interchangeably, see Lim [35] for the distinction between them.
[2]In this paper, these matrices are also called the $k$-Laplacian for simplicity.

**$k$-th homology vector space and embedding.** The homology vector space $\mathcal{H}_k$ is a subspace of $\mathcal{C}_k$ (loop space) such that every $k$-cycle (expressed as a $k$-cochain) in $\mathcal{H}_k$ is not the boundary of any $(k+1)$-cochain. In mathematical terms, $\mathcal{H}_k := \ker(\mathbf{A}_k)/\operatorname{im}(\mathbf{A}_{k+1})$. The rank of the subspace is called the $k$-th Betti number $\beta_k = \dim(\mathcal{H}_k)$, which counts the number of "loops" (*homology classes*) in the SC. $\mathcal{H}_k$ is equivalent to the null space of $\mathcal{L}_k$ [35, 48]; therefore, a basis of $\mathcal{H}_k$ can be obtained by the eigenvectors $\mathbf{Y} = [\mathbf{y}_1, \cdots, \mathbf{y}_{\beta_k}] \in \mathbb{R}^{n_k \times \beta_k}$ of $\mathcal{L}_k$ with eigenvalue 0. The *homology embedding* maps a $k$-simplex $\sigma_k$ to $\mathbf{Y}_{\sigma_k,:} = [\mathbf{y}_1(\sigma_k), \cdots \mathbf{y}_{\beta_k}(\sigma_k)]^\top \in \mathbb{R}^{\beta_k}$. Note that the basis $\mathbf{Y}$ is only identifiable up to a unitary transformation; hence, individual homology embedding coordinates might change with a different basis $\mathbf{Y}$.

**Continuous operators on manifolds.** The $k$-cochains are the discrete analogues of *$k$-forms* [59]. For $k = 1$, the following path integral [59] (along the geodesic $\gamma(t)$ connecting $x$ and $y$) relates a 1-cochain $\boldsymbol{\omega}$ to a 1-form $\mathbf{v}$ (vector field): $\boldsymbol{\omega}([x,y]) = \int_x^y \mathbf{v}(\gamma(t))\gamma'(t)\mathrm{d}t$. To estimate a vector field from $\boldsymbol{\omega}$, one can solve a least-squares problem [12], which is the inverse operation of the path integral (e.g., the vector fields in Figure 1 are estimated from $\mathbf{Y}$). Similarly, one can define the *differential* d and *codifferential* $\delta$ operators which are analog to $\mathbf{B}_{k+1}^\top$ and $\mathbf{B}_k$, respectively. The (continuous) $k$-Hodge Laplacian operators, which act on $k$-forms, can be defined for manifolds too, i.e., by $\Delta_k = \mathrm{d}_{k-1}\delta_k + \delta_{k+1}\mathrm{d}_k$. The homology group (the continuous version of $\mathcal{H}_k$) is defined as the null of $\Delta_k$. Its elements are harmonic $k$-forms $\zeta_k$, computed by solving $\Delta_k \zeta_k = 0$ with proper boundary conditions; they represent the continuous version of the discrete homology basis $\mathbf{Y}$.

**Connected sum and manifold (prime) decomposition.** The connected sum [33] of two $d$ dimensional manifolds $\mathcal{M} = \mathcal{M}_1 \sharp \mathcal{M}_2$ is built from removing two $d$ dimensional "disks" from each manifold $\mathcal{M}_1$, $\mathcal{M}_2$ and gluing together two manifolds at the boundaries (technical details in [33]). The analog of the connected sum for the abstract complexes will be defined in Section 3. The connected sum is a core operation in topology and is related to the concept of manifold (prime) decomposition. Informally speaking, the prime decomposition aims to factorize a manifold $\mathcal{M}$ into $\kappa$ smaller building blocks ($\mathcal{M} = \mathcal{M}_1 \sharp \cdots \sharp \mathcal{M}_\kappa$) so that each $\mathcal{M}_i$ cannot be further expressed as a connected sum of other manifolds. The well-known *classification theorem of surfaces* [3] states that any oriented and compact surface is the finite connected sum of manifolds homeomorphic to either a circle $\mathbb{S}^1$, a sphere $\mathbb{S}^2$, or a torus $\mathbb{T}^2$. Classification theorems for $d > 2$ are currently unknown; fortunately, the uniqueness of the prime decomposition for $d = 3$ (up to homeomorphism) was shown (Kneser-Milnor theorem [39]). Recently, Bokor et al. [9] (Corollary 2.5) showed the existence of factorizations of manifolds with $d \geq 5$, even though they might not be unique.

In this paper, we are interested in the following: given finite samples from $\mathcal{M}$, which is a $\kappa$-fold connected sum of $\mathcal{M}_i$, can this decomposition be recovered from the discrete homology embedding $\mathbf{Y}$ of $\mathcal{M}$? Namely, we would like to understand how $\mathbf{Y}$ relates to the homology embedding of each prime manifold $\mathcal{M}_i$.

## 3  Definitions, theoretical/algorithmic aims, and prior works

**Definitions.** We assume that the data $\mathbf{X}$ is sampled from a $d$-dimensional *oriented* manifold $\mathcal{M}$ that can be decomposed into $\kappa$ prime manifolds ($\mathcal{M} = \mathcal{M}_1 \sharp \cdots \sharp \mathcal{M}_\kappa$). Let $\mathcal{I}_i$ be an index set of the data points in $\mathbf{X}$ sampled from $\mathcal{M}_i$, for $i = 1, \ldots \kappa$. Denote by $\mathrm{SC}_k$, $\mathcal{L}_k$, $\mathcal{H}_k(\mathcal{M})$, and $\beta_k$ the simplicial complex, the $k$-Laplacian, the $k$-homology space, and the $k$-th Betti number of $\mathbf{X}$ sampled from $\mathcal{M}$. Furthermore, let $\widehat{\mathrm{SC}}_k^{(i)} = (\widehat{\Sigma}_0^{(i)}, \cdots, \widehat{\Sigma}_k^{(i)}), \widehat{\mathcal{L}}_k^{(ii)}, \mathcal{H}_k(\mathcal{M}_i), \beta_k(\mathcal{M}_i)$ be the same quantities for manifold $\mathcal{M}_i$ (supported on $\mathcal{I}_i$ for $i \leq \kappa$). $\widehat{\mathrm{SC}}_k$ and $\widehat{\mathcal{L}}_k$ (without superscript $i$) are the comparable notations for the disjoint manifolds $\mathcal{M}_i$'s, i.e. $\widehat{\mathrm{SC}} = \cup_{i=1}^\kappa \widehat{\mathrm{SC}}^{(i)} = (\widehat{\Sigma}_0, \cdots, \widehat{\Sigma}_k)$ with $\widehat{\Sigma}_\ell = \cup_{i=1}^\kappa \widehat{\Sigma}_\ell^{(i)}$ for $\ell \leq k$, and $\widehat{\mathcal{L}}_k$ is a block diagonal matrix with the $i$-th block being $\widehat{\mathcal{L}}_k^{(ii)}$. Additionally, let $\mathbf{Y}$ and $\widehat{\mathbf{Y}}$ (both in $\mathbb{R}^{n_k \times \beta_k}$) be the homology basis of $\mathcal{L}_k$ and $\widehat{\mathcal{L}}_k$, respectively. Let $\mathcal{S}_i$ be the index set of columns of $\widehat{\mathbf{Y}}$ corresponding to homology subspace $\mathcal{H}_k(\mathcal{M}_i)$, with $\mathcal{S}_i \cap \mathcal{S}_j = \emptyset$ for $i \neq j$, $|\mathcal{S}_i| = \beta_k(\mathcal{M}_i)$, and $\mathcal{S}_1 \cup \cdots \cup \mathcal{S}_\kappa = \{1, \cdots, \beta_k\}$. Since $\widehat{\mathbf{Y}}$ is the homology embedding of a block diagonal matrix $\widehat{\mathcal{L}}_k$, we choose it so that $[\widehat{\mathbf{Y}}]_{\sigma,m}$ equals the homology embedding of $\widehat{\mathcal{L}}_k^{(ii)}$ if

$\sigma \in \widehat{\Sigma}_k^{(i)}$ with column $m \in \mathcal{S}_i$ and is zero otherwise. Namely, $\widehat{\mathbf{Y}}$ lies in the direct sum of subspaces $\mathcal{H}_k(\mathcal{M}_i)$ for $i \leq \kappa$.

**Theoretical aim.** We are interested in the geometric properties of the null space eigenvectors $\mathbf{Y}$, and specifically in recovering the homology basis $\widehat{\mathbf{Y}}$ of the prime manifolds. Hence, we aim to bound the *distance* between the spaces spanned by $\mathbf{Y}$ and $\widehat{\mathbf{Y}}$. Under a small perturbation, one can provide an analogous argument to the *orthogonal cone* structure [40, 49] in spectral clustering (the zeroth homology embedding). The main technical challenge is that the connected sum of manifolds is a highly localized perturbation; namely, most cells are not affected at all, while those involved in the gluing process gain or lose $\mathcal{O}(1)$ (co)faces. Without properly designing $\mathcal{L}_k$ and $\widehat{\mathcal{L}}_k$, one might get a trivial bound.

**Algorithmic aim.** We exemplify the algorithmic aim using $k = 1$, $d = 2$, and $\kappa = 2$, particularly the genus-2 surface shown in Figure 1. The null space basis $\mathbf{Y}$ of $\mathcal{L}_k$ is only identifiable up to a unitary matrix due to the multiplicity of the zero eigenvalues. For instance, the top and bottom rows of Figure 1 are both valid bases for the harmonic edge flows in $\mathcal{H}_1$. However, the basis vector fields in the second row of Figure 1 are more inter-

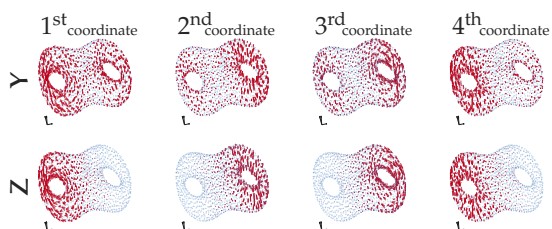

Figure 1: Harmonic vector fields obtained by solving a least-squares [12] with $\mathbf{Y}$ (top) and $\mathbf{Z}$ (bottom).

pretable than those in the top row because $\mathbf{Y}$ (the first row) is a linear combination of $\mathbf{Z}$ (the second row), with each basis (column in the figure) corresponding to a single homology class (loop). Therefore, here we propose a *data-driven* approach to obtain the optimal basis $\mathbf{Z}$ such that the coupling from other manifolds/subspaces is as weak as possible. Being able to obtain $\mathbf{Z}$ from an arbitrary $\mathbf{Y}$ can support numerous applications (more in Section 5); however, it is difficult to design a criterion for finding the optimal $\mathbf{Z}$ without knowing the geometric structure of the embedding of $\mathcal{H}_k$.

**Prior work.** The shape of the embedding by the principal eigenvectors of the graph Laplacian $\mathcal{L}_0$ is pivotal for showing the guarantees of spectral clustering algorithms for point cloud data or the inference algorithms for the stochastic block model. The analyses used either the matrix perturbation theory [40, 55, 56] or assume a mixture model [49]. For the higher-order $k$-Laplacian, it is reported empirically that the homology embedding is approximately distributed on the union (directed sum) of subspaces [21]; subspace clustering algorithms [30] were applied to partition edges/triangles under their framework.

On the application side, the eigen-embedding of $\mathcal{L}_k$ is used implicitly or explicitly in graph signal processing [27, 44, 45] and in learning algorithms utilizing Hodge decomposition [12, 28, 62]. They use the gradient/curl cochains (which correspond to the images of $\mathcal{L}_k^{\mathrm{down}}$ and $\mathcal{L}_k^{\mathrm{up}}$, respectively) in addition to the homology embedding ($\ker(\mathcal{L}_k)$); hence, our framework is intrinsically different from these works.

## 4 Main result: connected sum as a matrix perturbation

In this section, we analyze the geometric structure of $\mathbf{Y}$ by viewing the operation of *connected sum* through the lens of matrix perturbation theory [51]. We show that, under certain conditions, the homology embedding $\mathbf{Y}$ of the joint Laplacian $\mathcal{L}_k$ is approximated by $\widehat{\mathbf{Y}}$ for the simplices in $\Sigma_k \cap \widehat{\Sigma}_k$. In matrix terms, we show that $\mathbf{Y} \approx \widehat{\mathbf{Y}}\mathbf{O}$ (Theorem 1) with $\mathbf{O}$ a unitary transformation.

We first prepare our assumptions suited for SC built from point clouds. Most of the assumptions (except Assumption 1 for which the connected sum might not be defined) can be extended to the clique complex (for networks) or cubical complex (for images) without too many modifications.

**Assumption 1.** *The point cloud $\mathbf{X} \in \mathbb{R}^{n \times D}$ is sampled from a $d$-dimensional oriented and compact manifold $\mathcal{M} \subseteq \mathbb{R}^D$; the homology vector spaces $\mathcal{H}_k(\mathrm{SC})$ formed by the simplicial complex constructed from $\mathbf{X}$ are isomorphic to the homology group $\mathcal{H}_k(\mathcal{M})$ of $\mathcal{M}$, i.e., $\mathcal{H}_k(\mathrm{SC}) \simeq \mathcal{H}_k(\mathcal{M})$.*

*Furthermore, assume that $\mathcal{M} = \mathcal{M}_1 \sharp \cdots \sharp \mathcal{M}_\kappa$, and that $\mathcal{H}_k(\widehat{\mathrm{SC}}^{(i)}) \simeq \mathcal{H}_k(\mathcal{M}_i)$ for $i = 1, \cdots, \kappa$.*

This assumption is the minimal assumption needed for the analysis of the embedding of the $\mathcal{L}_k$; it states that any procedure to construct the simplicial complex or weight function for $\mathcal{L}_k$ is accepted as long as the isomorphic condition holds. The construction of the SC from the point cloud is out of the scope of this manuscript (see, e.g., Chen et al. [12] for building $\mathcal{L}_1$ from $\mathbf{X}$ with an analyzable limit and Latschev's theorem [32] on VR complexes). The last condition requires that the manifold $\mathcal{M}$ can be decomposed; this is most likely true, except for the known hard case of $\mathcal{M}$ with $d = 4$ discussed in Section 2. Note that Assumption 1 is similar to those used in the multi-manifold clustering [54], where they required the manifold to be constructed by a $\kappa$-fold union $\mathcal{M} = \mathcal{M}_1 \cup \cdots \cup \mathcal{M}_\kappa$. The goal of our framework is to analyze the homology embedding of $k$-Laplacian. By contrast, Trillos et al. [54] are interested in identifying each manifold with the notion of clustering ($\mathcal{H}_0$).

Assumption 1 is for points sampled from manifold $\mathcal{M}$ only. To make this assumption hold for networks or images, one can require that the $\mathcal{L}_k$ constructed from these two datasets can be roughly factorized into block-diagonal entries. Below we provide two other assumptions that are valid for both SC and CB (with some modifications): the first one controls the eigengap and the second one ensures a small perturbation in the spectral norm of $\mathcal{L}_k - \widehat{\mathcal{L}}_k$. By construction, $\mathcal{L}_k$ is positive semi-definite; since we are interested in the stability of its null space, we define, for any matrix $\mathbf{L} \succeq 0$, the *eigengap* as the smallest *non-zero* eigenvalue of $\mathbf{L}$ and denote it $\lambda_{\min}(\mathbf{L})$.

**Assumption 2.** *We denote the set of non-intersecting simplices be $\mathfrak{N}_k := \Sigma_k \cap \widehat{\Sigma}_k$. Let the set of destroyed and created $k$-simplices during connected sum by $\mathfrak{D}_k$ and $\mathfrak{C}_k$, respectively; they are defined by $\mathfrak{C}_k := \Sigma_k \backslash \mathfrak{N}_k$ and $\mathfrak{D}_k := \widehat{\Sigma}_k \backslash \mathfrak{N}_k$. We have: (1) no $k$-homology class is created during the connected sum process, i.e., $\beta_k(\mathrm{SC}) = \sum_{i=1}^{\kappa} \beta_k(\widehat{\mathrm{SC}}^{(i)})$. (2) The eigengaps of $\mathcal{L}_k^{\mathfrak{C},\mathfrak{C}}$ and $\widehat{\mathcal{L}}_k^{\mathfrak{D},\mathfrak{D}}$ are bounded away from the eigengaps of $\mathcal{L}_k^{(ii)}$, i.e., $\min\{\lambda_{\min}(\mathcal{L}_k^{\mathfrak{C},\mathfrak{C}}), \lambda_{\min}(\widehat{\mathcal{L}}_k^{\mathfrak{D},\mathfrak{D}})\} \gg \min\{\delta_1, \cdots, \delta_\kappa\}$, where $\delta_i$ is the eigengap of $\mathcal{L}_k^{(ii)}$.*

The first condition requires that the intersecting simplices $\mathfrak{D}_k \cup \mathfrak{C}_k$ do not create or destroy any $k$-th homology class; this holds, for instance, when the manifold $\mathcal{M}$ has dimension $d > k$. Under this condition, we have $\mathcal{H}_k(\mathcal{M}_1 \sharp \mathcal{M}_2) \simeq \mathcal{H}_k(\mathcal{M}_1) \oplus \mathcal{H}_k(\mathcal{M}_2)$ [33]. A counterexample for this condition is, e.g., inspecting the cavity space ($k = 2$) of a genus-2 surface built from gluing two tori together. That is, $\beta_2$ of a genus-2 surface is 1, while the sum of $\beta_2$ of two tori is 2. The second condition requires that the principal submatrix of $\mathcal{L}_k$ described by the block of $\mathfrak{C}_k \cup \mathfrak{D}_k$ has large eigengap. This happens, e.g., when $\mathfrak{C}_k$ and $\mathfrak{D}_k$ are cliques and are contained in small balls.

**Assumption 3** (Informal). *Let $\tilde{\mathbf{w}}_k = |\mathbf{B}_{k+1}[\mathfrak{N}_k, \mathfrak{N}_{k+1}]|\mathbf{w}_{k+1}$, $\tilde{\mathbf{w}}_{k-1} = |\mathbf{B}_k[:, \mathfrak{N}_k]|\tilde{\mathbf{w}}_k$. For $\ell = k$ or $k - 1$, we have $\max_{\sigma \in \mathfrak{N}_\ell} \{w_\ell(\sigma)/\tilde{w}_\ell(\sigma) - 1\} \leq \epsilon_\ell$, $\max_{\sigma \in \mathfrak{N}_\ell} \{\widehat{w}_\ell(\sigma)/\tilde{w}_\ell(\sigma) - 1\} \leq \epsilon_\ell$, and $\max_{\sigma \in \mathfrak{N}_\ell} \{|w_\ell(\sigma)/\widehat{w}_\ell(\sigma) - 1|\} \leq \epsilon'_\ell$. Assumption S1 is the formal version of this assumption.*

For $k = 1$, it states that not too many triangles are being created or destroyed during connected sum. For this assumption to hold, the sampling density in the connected sum region should be smaller than in other regions, i.e., the manifold $\mathcal{M}$ should be sparsely connected (e.g., Figure 2a). Empirically, we observed that the perturbation is small even when $\mathcal{M}$ is not sparsely connected (more discussions in Section 6). Note also that $\epsilon'_\ell \ll \epsilon_\ell$, for $\epsilon'_\ell$ represents the *net* change in the degree after connected sum. It might be possible to obtain a tighter bound fully by $\epsilon'_\ell$'s, which do not depend on the relative density between the connected sum region and the remaining manifolds; we leave it as future work.

**Theorem 1.** *Let $\mathsf{DiffL}_k^{\mathrm{down}}$ ($\mathsf{DiffL}_k^{\mathrm{up}}$) be the modified difference (defined in Supplement A) of $\mathcal{L}_k^{\mathrm{down}}$ and $\widehat{\mathcal{L}}_k^{\mathrm{down}}$ (respectively $\mathcal{L}_k^{\mathrm{up}}$ and $\widehat{\mathcal{L}}_k^{\mathrm{up}}$). Under Assumptions 1–3 with notations defined as before and $\lambda_k = k + 2$, if $\left\|\mathsf{DiffL}_k^{\mathrm{down}}\right\|^2 \leq \left[2\sqrt{\epsilon'_k} + \epsilon'_k + \left(1 + \sqrt{\epsilon'_k}\right)^2 \sqrt{\epsilon'_{k-1}} + 4\sqrt{\epsilon_{k-1}}\right]^2 \lambda_{k-1}^2$ and $\|\mathsf{DiffL}_k^{\mathrm{up}}\|^2 \leq \left[2\sqrt{\epsilon'_k} + \epsilon'_k + 2\epsilon_k + 4\sqrt{\epsilon_k}\right]^2 \lambda_k^2$, then there exists a unitary matrix $\mathbf{O} \in \mathbb{R}^{\beta_k \times \beta_k}$ such that*

$$\left\|\mathbf{Y}_{\mathfrak{N}_k,:} - \widehat{\mathbf{Y}}_{\mathfrak{N}_k,:}\mathbf{O}\right\|_F^2 \leq \frac{8\beta_k \left[\left\|\mathsf{DiffL}_k^{\mathrm{down}}\right\|^2 + \|\mathsf{DiffL}_k^{\mathrm{up}}\|^2\right]}{\min\{\delta_1, \cdots, \delta_\kappa\}}. \tag{2}$$

The proof (in Supplement A) is based on bounding the error between $\mathcal{L}_k$ and $\widehat{\mathcal{L}}_k$ with $\tilde{\mathcal{L}}_k$ (the Laplacian after removal of $k$-simplices during connected sum), the use of a variant of the Davis-

Kahan theorem [61], and the bound of the spectral norm of $\mathcal{L}_k$ for a simplicial complex, i.e., $\|\mathcal{L}_k\|_2 \leq \lambda_k = k + 2$ [26].

The LHS of (2) contains only the simplices in $\mathfrak{N}_k$ because $\mathbf{Y}$ is not defined on $\mathfrak{D}_k$ (similar for $\widehat{\mathbf{Y}}$ and $\mathfrak{C}_k$). Nonetheless, (2) makes sure that the (unbounded) perturbations in the embedding of $\mathfrak{C}_k \cup \mathfrak{D}_k$ do not propagate to the rest of the simplices. The bound in (2) can be extended to CB (Corollary 2) by changing the $\lambda_k$ value from $(k+2)$ to $2k+2$. The $2k+2$ term here is the maximum eigenvalue of the $\mathcal{L}_k$ built from any cubical complex (Proposition S3).

**Corollary 2** (For $\mathcal{L}_k$ built from a CB). *Under Assumptions 2–3 with* $\mathsf{DiffL}_k^{\mathrm{up}}$ *as well as* $\mathsf{DiffL}_k^{\mathrm{down}}$ *defined in Theorem 1 and* $\lambda_k = 2k + 2$, *there exists a unitary matrix* $\mathbf{O}$ *such that* (2) *holds.*

---

**Algorithm 1:** Subspace identification

**Input** : SC, $k$, weights $\mathbf{W}_{k+1}$
1   $\mathbf{B}_k, \mathbf{B}_{k+1} = \text{BOUNDARYMAPS}(\text{SC}, k)$
     ▷ in Algorithm S2
2   **for** $\ell = k, k-1$ **do**
3     $\mathbf{W}_\ell \leftarrow \text{diag}\{|\mathbf{B}_{\ell+1}|\mathbf{W}_{\ell+1}\mathbf{1}_{n_{\ell+1}}\}$
4     $\mathbf{A}_{\ell+1} \leftarrow \mathbf{W}_\ell^{-1/2}\mathbf{B}_{\ell+1}\mathbf{W}_{\ell+1}^{1/2}$
5   $\mathcal{L}_k = \mathbf{A}_k^\top \mathbf{A}_k + \mathbf{A}_{k+1}\mathbf{A}_{k+1}^\top$
6   $\mathbf{Y} \in \mathbb{R}^{n_k \times \beta_k} \leftarrow \text{NULLSPACE}(\mathcal{L}_k)$
7   $\mathbf{Z} \leftarrow \text{ICANOPREWHITE}(\mathbf{Y})$
**Return** : Independent basis $\mathbf{Z}$

---

**Subspace $\mathcal{H}_k(\mathcal{M}_i)$ identification.** We propose to approximately recover $\widehat{\mathbf{Y}}$ from the columns of the coupled basis $\mathbf{Y}$ by *blind source separation*, as described by Algorithm 1. Specifically, the *independent basis* $\mathbf{Z}$ is obtained by Infomax ICA [6] on $\mathbf{Y}$ of $\mathcal{L}_k$, with a modification (Line 7) that preserves the necessary properties of harmonic cochains (i.e., they are *divergence-free* and *curl-free*, see also Proposition 3). Algorithm 1 works for CB as well by using the appropriate $\mathbf{B}_k$, $\mathbf{B}_{k+1}$ construction method (Line 1). Each column of the obtained $\mathbf{Z}$ is now approximately supported on a single prime manifold $\mathcal{M}_i$.

## 5   Applications: homologous loops detection, clustering, and visualization

**Homologous loop detection.** In addition to the *rank* information, one might find it beneficial to extract the shortest cycle of the corresponding $\mathcal{H}_k$ generator. This application is found useful in domains including finding minimum energy trajectories in molecular dynamics datasets, trajectory inference in RNA single-cell sequencing [46], and segmenting circular structures in medical images [50]. We propose a *spectral* shortest homologous loop detection algorithm (Algorithm 2) based on the shortest path algorithm (Dijkstra) as follows: for each dimension $i = 1, \cdots, \beta_1$, the algorithm reverses every edge $e$ having negative $[\mathbf{z}_i]_e$ to generate a weighted digraph $G_i = (V, E_i)$ (Lines 2–4), with the weight of edge $e = (j, j') \in E_i$ equal to the

---

**Algorithm 2:** Spectral homologous loop detection

**Input** : $\mathbf{Z} = [\mathbf{z}_1, \cdots, \mathbf{z}_{\beta_1}]$, $V$, $E$, edge distance $\mathbf{d}$
1   **for** $i = 1, \cdots, \beta_1$ **do**
2    $E_i^+ \leftarrow \{(s,t) : (s,t) \in E \text{ and } [\mathbf{z}_i]_{(s,t)} > 0\}$
3    $E_i^- \leftarrow \{(t,s) : (s,t) \in E \text{ and } [\mathbf{z}_i]_{(s,t)} < 0\}$
4    $\tau \leftarrow \text{PERCENTILE}(|\mathbf{z}_i|, 1 - 1/\beta_1)$
5    $E_i^\times \leftarrow \{e \in E_i^+ \cup E_i^- : |[\mathbf{z}_i]_e| < \tau\}$
6    $E_i \leftarrow E_i^+ \cup E_i^- \backslash E_i^\times$
7    $G_i \leftarrow (V, E_i)$, with weight of $e \in E_i$ being $[\mathbf{d}]_e$
8    $d_{\min} = \texttt{inf}$
9    **for** $e = (t, s_0) \in E_i$ **do**
10     $\mathcal{P}^*, d^* \leftarrow \text{DIJKSTRA}(G_i, \texttt{from=}s_0, \texttt{to=}t)$
      ▷ Note that $\mathcal{P}^* = [s_0, s_1, \cdots, t]$
11     **if** $d^* < d_{\min}$ **then**
12      $\mathcal{C}_i \leftarrow [t, s_0, s_1, \cdots, t]$

**Return** : $\mathcal{C}_1, \cdots, \mathcal{C}_{\beta_1}$

---

Euclidean distance $[\mathbf{d}]_{(j,j')} = \|\mathbf{x}_j - \mathbf{x}_{j'}\|_2$. The algorithm finds a shortest (in terms of $\mathbf{d}$) loop on this weighted digraph for each $i$ and outputs it as the homologous loop representing the $i$-th class. We present the following proposition (with the proof in Supplement B) to support Algorithm 2; it implies that if each coordinate of $\mathbf{Z}$ extracted from Algorithm 1 corresponds to a homology class, then the detected homologous loop for each homology class is the shortest.

**Proposition 3.** *Let $\mathbf{z}_i$ for $i = 1, \cdots, \beta_1$ be the $i$-th homology basis that corresponds to the $i$-th homology class. For every $i = 1, \cdots, \beta_1$, (1) there exist at least one* cycle *in the digraph $G_i$ such that every vertex $v \in V$ can traverse back to itself (*reachable*); (2) the corresponding* cycle *will enclose at least one homology class (*no short-circuiting*).*

Since every vertex is *reachable* from itself, we are guaranteed to find a loop for any starting/ending pair (Lines 9–12). Additionally, there will be no short-circuiting for any loop; each loop we found from Dijkstra is guaranteed to be non-trivial. However, there is one caveat from the second property: even though the $i$-th loop is non-trivial, it might not always be corresponding to the $i$-th homology class due to the noise in small $[\mathbf{z}_i]_e$. Namely, loops that do not represent $i$-th homology class can be formed with edges $e$ having small $[\mathbf{z}_i]_e$, resulting in the instability and the (possible) duplication of the identified loops. To address the issue, we propose a heuristic thresholding, by which we keep the $n_1/\beta_1$ edges with the largest absolute value in $|[\mathbf{z}_i]_e|$ (Lines 4–5). We chose to keep $n_1/\beta_1$ by treating each homology class equally, i.e., each class has roughly $n_1/\beta_1$ edges. Unlike Theorem 1 or Algorithm 1, Algorithm 2 cannot be extended to the case when $k \geq 2$ because the Dijkstra algorithm is employed. We leave its generalization to extract higher-order cavities as future work.

Compared with the previous approach that finds the shortest loops [16] combinatorially, our approach has better time complexity; specifically, the algorithm by Dey et al. [16] has time complexity $\mathcal{O}(nn_1^3 + nn_1^2 n_2)$, whereas Algorithm 2 runs in time $\mathcal{O}(n_1^{2.37\cdots} + \beta_1^2 n_1 + \beta_1 n_1 n \log n)$. The first, second, and third terms correspond to the time complexity of eigendecomposition of $\mathcal{L}_1$, the Info-max ICA, and the Dijkstra algorithm on every digraph $G_i$, respectively. Note that if the simplicial complex is built from point clouds, the number of triangles $n_2$ may be large; this dependency on $n_2$ makes the algorithm [16] hard to scale. On the other hand, our framework requires that $\mathbf{z}_i$ are each supported on one homology class; therefore, loops can only be correctly identified using Algorithm 2 if the manifold is sparsely connected (Assumptions 1–3). Additional comparisons between our algorithm and other methods that pose special requirements on the analyzed data can be found in Supplement D.4.2.

**Classifying any 2-dimensional manifold.** The Betti number $\beta_1$ of a torus is 2, which is equal to that of two disjoint circles; hence one cannot distinguish these two manifolds *only* by rank information. Fortunately, they can be categorized using the homology embedding $\mathbf{Z}$. By the classification theorem [3], any 2D surface is the connected sum of circles $\mathbb{S}^1$ and tori $\mathbb{T}^1$; therefore, Theorem 1 indicates that embedding lies approximately in the directed sum of homology subspace of $\mathbb{S}^1$ and/or $\mathbb{T}^2$. The homology embedding of $\mathbb{S}^1$ is a line since it is in $\mathbb{R}^1$. On the other hand, any loop in a torus can be a convex combination of the two homology classes, implying that the intrinsic dimension of the homology embedding is 2. It is hard to obtain $\mathbf{Z}$ of any arbitrary torus; we present the homology embedding of the flat $m$-torus below by expressing the null space basis (1-cochains) as the path integrals of the corresponding harmonic 1-forms [12, 59].

**Proposition 4.** *The envelope of the first homology embedding (1-cochain) induced by the harmonic 1-form on the flat m-torus $\mathbb{T}^m$ is an $m$-dimensional ellipsoid.*

The proof (in Supplement B) is straightforward thus is omitted here. Proposition 4 and the classification theorem suggest that the first homology embedding is either a line, a disk, or a combination of the two (with replacement). See an example for the genus-2 surface in Figures 2j and S1.

Note that it is possible to classify any 2-manifold with higher-dimensional homology groups; for instance, one can distinguish $\mathbb{T}^2$ from $\mathbb{S}^1 \sharp \mathbb{S}^1$ by the second homology group. However, our approach scales better in computation or memory usage since inspecting $\mathcal{H}_2$ needs at least the calculation and storage of tetrahedrons from the neighborhood graph.

**Other applications.** As pointed out earlier, one can visualize the basis of the harmonic vector fields (of $\mathcal{H}_k$) by overlaying the columns of $\mathbf{Y}$ onto the original dataset (Figure 1). Being able to successfully extract a decoupled basis $\mathbf{Z}$ increases the interpretability of $\mathcal{H}_k$, as shown in the second row of Figure 1. Theorem 1 also supports the use of subspace clustering algorithm in the higher-order simplex clustering framework [21].

# 6   Experiments

We demonstrate our approach by computing $\mathbf{Y}, \mathbf{Z}$ and the shortest loops for five synthetic manifolds: two of them are prime manifolds (TORUS *torus*, 3-TORUS *three-torus*) and three (PUNCTPLANE *punctured plane with two holes*, GENUS-2 *genus-2*, and TORI-CONCAT *concatenation of 4 tori*) are factorizable manifolds. Furthermore, five additional real point clouds (ETH and MDA from chemistry, PANCREAS from biology, 3D-GRAPH from 3D modeling, and ISLAND from oceanography) are ana-

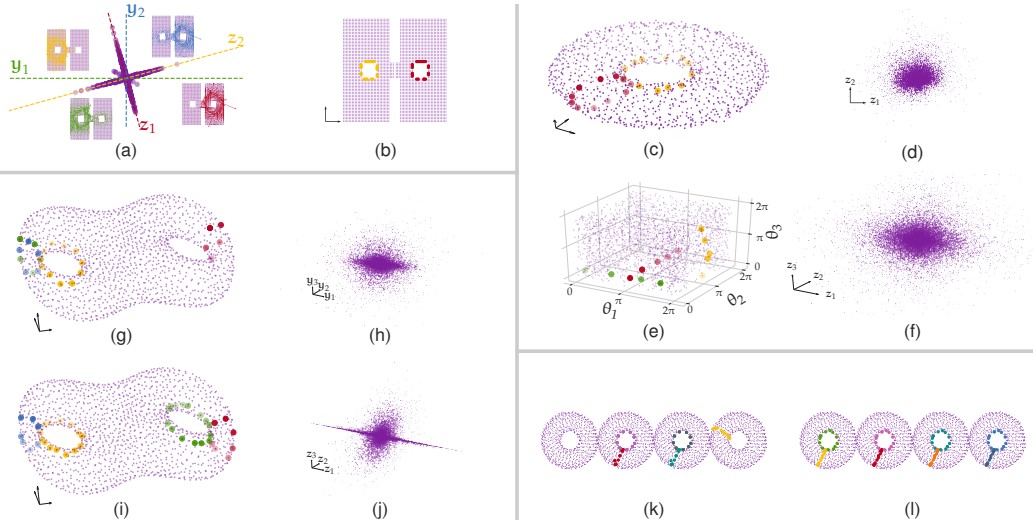

Figure 2: (a) The first homology embedding of PUNCTPLANE. The harmonic vector fields are overlaid on the data in the inset plots; green, blue, red, and yellow arrows correspond to $\mathbf{y}_1$, $\mathbf{y}_2$, $\mathbf{z}_1$, and $\mathbf{z}_2$, respectively. (b), (c), (e), (i), and (l) are the detected loops using Dijkstra on $\mathbf{Z}$ for PUNCTPLANE (colors are in (a)), TORUS, 3-TORUS, GENUS-2, and TORI-CONCAT, respectively. (g) and (k) represent the identified loops on the coupled embedding $\mathbf{Y}$ for GENUS-2 and TORI-CONCAT, respectively. (d), (f), (h), and (j) present the embeddings used to detect loops in (c), (e), (g), and (i), respectively.

lyzed under this framework. For all the point clouds, we build the VR complex SC from the CkNN kernel [8] so that the resulting $\mathcal{L}_1$ is sparse and the topological information is preserved. Note that other methods for building an SC from $\mathbf{X}$ can also be used as long as $\mathcal{H}_k$ is successfully identified (Assumption 1). Lastly, we illustrate the efficacy of our framework to a non-manifold data: RETINA from medical imaging. Please refer to Supplement D for detailed discussions on procedures to generate, preprocess, and download these datasets. All experiments are replicated more than five times with similar results. We perform our analysis on a desktop running Linux with 32GB RAM and an 8-Core 4.20GHz Intel® Core™ i7-7700K CPU; every experiment completes within 3 minutes (1-2 minutes on eigendecomposition of $\mathcal{L}_1$, and around 30 seconds on both ICA and Algorithm 2).

**Synthetic manifolds.** The results for the synthetic manifolds are in Figure 2. Figure 2a (the harmonic embedding of PUNCTPLANE) confirms Theorem 1 that $\mathbf{Y}$ is approximately distributed on two subspaces (yellow and red), with each loop parametrizing a single hole (inset of Figure 2a). As discussed previously in Figure 1, the harmonic vector bases (green and blue) are mixtures of the separate subspaces; therefore, these bases have poor interpretability compared with the independent subspace $\mathbf{Z}$ identified by Algorithm 1. The shortest loops (Figure 2b) corresponding to $\mathbf{z}_1$ (yellow), $\mathbf{z}_2$ (red) are obtained by running Dijkstra on the digraphs induced by $\mathbf{z}_1$ and $\mathbf{z}_2$ separately (Algorithm 2). Figures 2c–2f show the results of the two simple *prime manifolds*: TORUS and 3-TORUS. The harmonic embeddings of TORUS (Figure 2d) and 3-TORUS (Figure 2f) are a two-dimensional disk and a three-dimensional ellipsoid, respectively; this confirms the conclusion from Proposition 4. The shortest loops obtained from Algorithm 2 for these two datasets are in Figures 2c and 2e, showing that these loops travel around the holes in TORUS (or 3-TORUS). Note that we plot 3-TORUS in the intrinsic coordinate because a three torus can not be embedded in 3D without breaking neighborhood relationships. Three lines in 2e are indeed loops due to the periodic boundary condition, i.e., $0 = 2\pi$, in the intrinsic coordinate. Figures 2h and 2j show the embedding of the coupled harmonic basis ($\mathbf{Y}$) and that corresponding to the independent subspace ($\mathbf{Z}$) obtained by Algorithm 1. Compared with $\mathbf{Y}$, each coordinate of $\mathbf{Z}$ corresponds to a subspace, i.e., the left or right handle of GENUS-2, and does not couple with other homology generators. $\mathbf{Z}$ is thus a union of two 2D disks, with each disk approximating the harmonic embedding of a torus (see Figure S1 for more detail). Compared with the loops obtained by running Algorithm 2 on $\mathbf{Y}$ (Figure 2g), each loop in Figure 2i identified from $\mathbf{Z}$ parameterizes the corresponding homology generator without being homologous to other loops. Similar results on TORI-CONCAT are in Figures 2k and 2l, which correspond to the

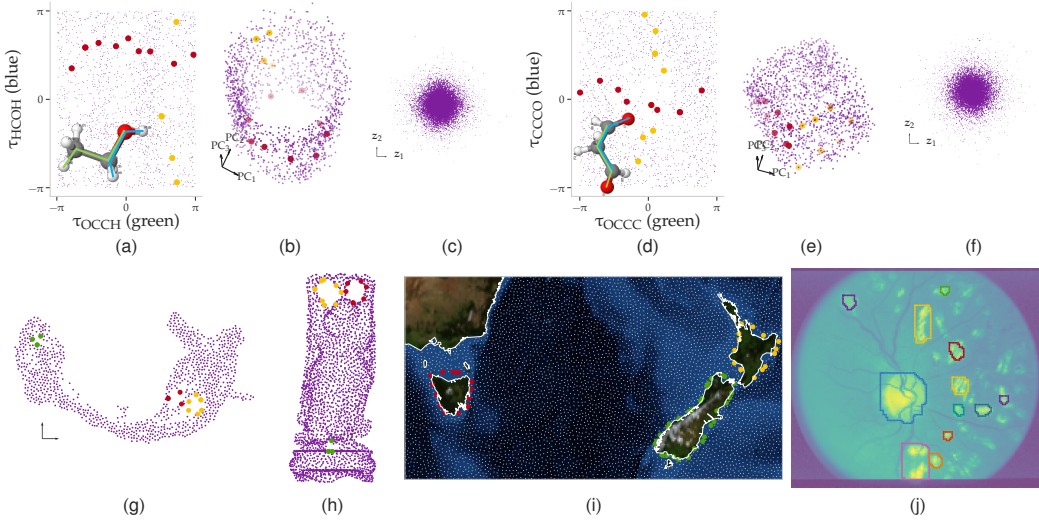

Figure 3: (a) and (b) are the detected loops of `ETH` using Dijkstra on $\mathbf{Z}$ (in (c)) in the torsion space (inset of (a)) and in the PCA space, respectively. (d)–(f) are the results for `MDA` that are similar to those for `ETH` in (a)–(c). (g)–(j) show the identified loops using $\mathbf{Z}$ for `PANCREAS`, `3D-GRAPH`, `ISLAND`, and `RETINA`, respectively.

loops obtained from $\mathbf{Y}$ and $\mathbf{Z}$, respectively. The pairwise scatter plots of the eight-dimensional $\mathbf{Z}$ (or $\mathbf{Y}$) are in Figure S2 of Supplement D. Note that `PUNCTPLANE` is an example of a sparsely connected manifold (see the low-density area in the middle), with $\epsilon_1 \approx 0.035$ and $\epsilon_0 \approx 0.038$. Manifolds of other synthetic/real datasets might not be sparsely connected due to the (approximately) constant sampling densities; nevertheless, the perturbations to the subspaces remain small for these datasets.

**Small molecule data [13].** Figures 3a–3c and 3d–3f show our analysis on `ETH` and `MDA`, respectively. These two small molecule datasets, whose ambient dimensions are $D = 102$ and $D = 98$, are suggested to be noisy non-uniformly sampled tori [52]; the harmonic embeddings of these two datasets (Figures 3c and 3f ) confirm this idea. Finding the minimum trajectories corresponding to a specific bond torsion is of interest in chemistry; in these two molecular dynamics systems, this problem can be translated into finding the homologous loops in the point cloud. The homologous loops found by Algorithm 2 overlaid on the first three *principal components* (PCs) for these two datasets can be found in Figures 3b (for `ETH`) and 3e (for `MDA`). The same homologous loops plotted in the bond torsion space (with definition in the insets) based on our prior knowledge are in Figures 3a and 3d. Similar to the discussion for `3-TORUS` (Figure 2e), the yellow/red trajectories form loops due to the periodic boundary condition of the bond torsions.

**RNA single-cell sequencing data [7].** The *trajectory inference* methods [46] for analyzing the RNA single-cell sequencing datasets aim to order the cells (points in high-dimensional expression space) along developmental trajectories, which are inferred from the structure of the point clouds. Identifying loops in the dataset can serve as a building block for delineating a correct trajectory, especially for determining cell cycle and cell differentiation. To illustrate the idea, we compute the 1-Laplacian on the CkNN kernel [8] constructed on the UMAP [36] embedding (Algorithm 1). Figure 3g shows the identified loops from Algorithm 2, with the green loop being the cycle of ductal cells and yellow/red loops representing a trifurcation (endocrine cell differentiation).

**Additional point cloud datasets.** `3D-GRAPH` [15] is a 3D model of a Buddha statue with a pre-computed triangulation. We treat the 3D model as a point cloud and subsample 3000 farthest points from the original dataset; $\mathcal{L}_1$ is obtained from the VR complex of the CkNN kernel. Note that with this small sample size, two smaller loops near the waist of the statue are not detectable. Hence, the number of zero eigenvalues of $\mathcal{L}_1$ is 3, with the corresponding homology generators shown in Figure 3h. `ISLAND` [23], which contains ocean buoys around the Tasman sea, is the other point cloud in

our analysis. The estimated $\beta_1$ is 3, with the detected loops being the North Island of New Zealand, the South Island of New Zealand, and the main island of Tasmania (Figure 3i).

**Non-manifold dataset.** Our framework for identifying subspaces is still valid for cubical complexes built from images (by Corollary 2). We demonstrate the idea on `RETINA`, a medical retinal image [25]. The cubical complex is constructed by intensity thresholding (also called the sub-level set method in TDA [58]) and then applying morphological closing on the binary image to remove small cavities. The weight for every rectangle $\mathbf{w}_2(\sigma)$ is set to 1; the estimated null space dimension of the $\mathcal{L}_1$ built from CB is $\beta_1 = 12$, with the identified homologous loops in Figure 3j. The result shows the robustness of the proposed framework even for large $\beta_1$.

# 7    Conclusion and broader impact

Our contributions in the emerging field of spectral algorithms for $k$-Laplacians $\mathcal{L}_k$ [12, 21, 35, 48] are summarized as follows. (i) We extend the study of the homology embedding of vertices by the graph Laplacian $\mathcal{L}_0$ (spectral clustering) to those of higher-order simplices by $\mathcal{L}_k$. Specifically, the $k$-th homology embedding can be approximately factorized into parts, with each corresponding to a prime manifold given a small perturbation (small $\epsilon_\ell$ and $\epsilon'_\ell$ for $\ell = k, k-1$). (ii) The analysis is made possible by expressing the $\kappa$-fold *connected sum* as a matrix perturbation. This convenient property of the homology embedding supports (iii) the use of ICA to identify each decoupled subspace and motivates (iv) the application to the shortest homologous loop detection problem. The proposed framework opens up numerous future directions for us to explore. For instance, one can extend our framework to investigate the generalization of the loosely connected clusters, i.e., when the null space $\mathcal{H}_k$ eigenvalues are not strictly zero; it applies to the case when analyzing the noisy topological structures in the data manifold. Moreover, the connections of our framework to the recent advances in spectral TDA and representation learning can be further explored; they include the persistent spectral method [38, 57] and the disentanglement of representation [63] in generative modeling.

**Broader impact.** Our analysis provides insight into the structure of the $k$-th harmonic embedding. This framework can inspire researchers in developing *spectral* topological data analysis algorithms (e.g., visualization, clustering, tightest higher-order *cycles* for $k \geq 2$ [20, 42]) similar to those that were inaugurated by spectral clustering two decades ago. These applications are especially beneficial to scientists (chemists, biologists, oceanographers, etc.) who use high-dimensional data analysis techniques for studying complex systems. Similar to the limitation of other unsupervised learning algorithms, practitioners without solid understandings of *both* the analyzed datasets and the used algorithm might draw controversial conclusions (see, e.g., discussions in [2, 41]). Possible approaches to mitigate the negative consequences are to design proper validation and causal inference algorithms for this framework; we leave them as potential directions we will explore.

# Acknowledgements

The authors acknowledge partial support from the U.S. Department of Energy's Office of Energy Efficiency and Renewable Energy (EERE) under the Solar Energy Technologies Office Award Number DE-EE0008563 and from the National Science Foundation award DMS 2015272. They thank Alexandre Tkatchenko, Jim Pfaendtner, Stefan Chmiela, and Chris Fu for providing the molecular dynamics data as well as for many hours of brainstorming and advice.

# Disclaimer

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
