# OpenReview forum: "The decomposition of the higher-order homology embedding constructed from the $k$-Laplacian"
_NeurIPS.cc/2021/Conference — NeurIPS 2021 Oral_

### Official Review · Reviewer_R3gu · 2021-07-02

**Rating:** 9
**Confidence:** 5

**Summary:**

In this manuscript, the authors study the decomposition of the homology spaces from Hodge Laplacian matrices, and propose a new algorithm for the detection of the shortest homology generators. In their model, the direct sum of homology embedding or spaces from Hodge Laplacian is approximated by connected sum of manifolds. More specifically, the homology space is decomposed into subspaces with simple topological components. An algorithm is designed to detect shortest homology loops in the structure. The approach to use homology spaces for clustering and detection of shortest loops is very novel and interesting. It expands the area of topological data analysis, and will have a great impact for various data modelling and analysis.

**Ethical Concerns:**

None.

**Main Review:**

Major problems,

1)	The claim that “Analysis of the eigenfunctions of H0 of the graph Laplacian L0 is pivotal in providing guarantees for spectral clustering and community detection algorithms” is not proper at all! In MOST (not all) spectral clustering and community detection problems, their graphs are well-connected and the corresponding L0 matrices have only one zero-eigenvalue (with a trivial eigenvector). In these models, the clustering and community are determined by the eigenvectors from the smallest non-zero eigenvalues. Note that only in some special cases, graphs are not well-connected and separated into individual subgraphs and  “eigenfunctions of H0” is “pivotal”!

2)	The authors need to provide more tests and comparisons to support their claim that “Our spectral loop detection algorithm scales better than existing methods and is effective on diverse data such as point clouds and images”. In fact, “Shortest homologous loop” is a very interesting and important topic that has been actively studied in computational geometry/topology. There are a lot of existing works in this area. To name only a few,

Feng, Xin, and Yiying Tong. "Choking loops on surfaces." IEEE transactions on visualization and computer graphics 19.8 (2013): 1298-1306.

Dey, Tamal K., Fengtao Fan, and Yusu Wang. "An efficient computation of handle and tunnel loops via Reeb graphs." ACM Transactions on Graphics (TOG) 32.4 (2013): 1-10.

Chambers, Erin W., Jeff Erickson, and Amir Nayyeri. "Minimum cuts and shortest homologous cycles." Proceedings of the twenty-fifth annual symposium on Computational geometry. 2009.

Dey, Tamal K., Tao Hou, and Sayan Mandal. "Persistent 1-cycles: Definition, computation, and its application." International Workshop on Computational Topology in Image Context. Springer, Cham, 2019.

Chao Chen, Daniel Freedman: "Hardness results for homology localization", in Proceedings of the Twenty-First Annual ACM-SIAM Symposium on Discrete Algorithms (SODA), 2010

T. K. Dey, A. Hirani, and B. Krishnamoorthy.   "Optimal homologous cycles, total unimodularity, and linear programming." SIAM J. Computing, Vol. 40, pp 1026--1044. Prelim. version in 42nd ACM Sympos. Comput. Theory (STOC 2010)

Oleksiy Busaryev, Sergio Cabello, Chao Chen, Tamal K. Dey, Yusu Wang: "Annotating simplices with a homology basis and its applications", in Scandinavian Symposium and Workshops on Algorithm Theory (SWAT), 2012

Pengxiang Wu, Chao Chen, Yusu Wang, Shaoting Zhang, Changhe Yuan, Zhen Qian, Dimitris Metaxas, Leon Axel:  "Optimal Topological Cycles and Their Application in Cardiac Trabeculae Restoration", in Information Processing in Medical Imaging (IPMI), 2017

In this manuscript, the authors only compare their model with the one in Ref 15 (even though they have mentioned Ref 16 in their paper) and claim that their models “scales better than existing methods”. The authors are suggested to add more discussions and comparisons to these existing models.

3)	In fact, homology spaces and their applications have been extensively studied through Hodge decomposition.  To name only a few,

Jiang, Xiaoye, Lek-Heng Lim, Yuan Yao, and Yinyu Ye. "Statistical ranking and combinatorial Hodge theory." Mathematical Programming 127, no. 1 (2011): 203-244.

Zhao R, Wang M, Chen J, Tong Y, Wei GW. The de Rham–Hodge Analysis and Modeling of Biomolecules. Bulletin of mathematical biology. 2020 Aug;82(8):1-38.

The authors are suggested to add more discussions about these progresses.

Minor problems,
1)	The term “k-Laplacian” used in this paper is not very proper. It should be k-th combinatorial Laplacian or Hodge Laplacian. There are various definitions of Laplacian matrixes, such as Graph Laplacian, Hodge Laplacian, Tarski Laplacian, path Laplacian, etc. The authors need to be more precise to avoid confusion.

2)	Page 2, line 64, typo in the equation for boundary matrix.

3)	Page 3, line 86, only $y_1$ eigenvector is used in the expression? I guess it should be a typo.

4)	Recently, there are great advancements in persistent spectral models from Hodge Laplacian. With persistent eigen-information (both eigenvalue and eigenvectors), Persistent spectral based machine learning models outperform ALL existing machine learning models (>40) in common benchmark dataset, demonstrating the great advantage and significance of Hodge Laplacian models in real application!

Rui Wang, Duc D Nguyen and Guo-Wei Wei, Persistent spectral graph, International Journal for Numerical Methods in Biomedical Engineering, 36(9), e3376  (2020).

Zhenyu Meng and Kelin Xia, "Persistent spectral based machine learning (PerSpect ML) for protein-ligand binding affinity prediction", Science advances, 7(19), eabc5329, 2021


**Time Spent Reviewing:**

20

---

> ### Author Response · Authors · 2021-08-10
> **Authors' Response to Reviewer 4 (R3gu)**
>
> Thanks for all the constructive feedback on our paper! We will make sure
> to include the discussions (the null space of
> ${\mathbf{\mathcal L}}_0$, existing works of "shortest homologous
> loop" algorithms, the applications using Hodge decomposition, and the
> "persistent spectral based ML" methods) in our final version of the
> paper. We will also fix the typos and improve our final version based on
> the suggestions provided.
>
> 1.  ***"Null space of L0 not "pivotal":"*** We heartily agree!! We will
>     amend to "pivotal for the study of clusterings/SBMs with perfect
>     separations" or remove. Indeed, the analogy of the spectral
>     clustering on the well-connected graph to the higher-order
>     ${\mathbf{\mathcal L}}_k$ is an interesting extension to this
>     work; specifically, one might be able to study the cases when
>     ${\mathbf{\mathcal L}}_k$ has "interesting" eigenfunctions for
>     small but non-zero eigenvalues.
>
> 2.  ***"The authors are suggested to add more discussions and
>     comparisons to these existing models."*** Thank you for the many
>     references. We will discuss these approaches in the final version of
>     the paper and summarize some of them as follows. We note that these
>     related works offer efficient and elegant alternatives in cases when
>     some extra information is available, e.g., a triangular mesh, having
>     an initial non-trivial loop, or in the special case of surfaces in
>     3D. Thus, we see our work, which does apply to point clouds/VR
>     complexes with any ambient or intrinsic dimension, as complementary
>     to these methods. Nevertheless, a real-time comparison on surfaces,
>     possibly with noise, would be highly informative and we plan to
>     perform it. **Thanks again for bringing these references to our
>     attention.**
>
>     1.  \[DHK11\] proposed an integer programming formulation
>         of finding the optimal homology loops given a non-trivial cycle;
>         this problem is intrinsically different from our spectral loops
>         extraction algorithm since no loops are given up-front.
>         Nevertheless, the integer programming problem can be relaxed to
>         linear programming when the boundary matrix
>         ${{\mathbf{B}}}_2$ is total unimodular, which can be solved
>         in $\mathcal O^*(\sqrt{n_1}n_2 + n_1^{5/2})$ by \[LS14\].
>         An example of the total unimodularity assumption to hold is when
>         $\mathrm{SC}$ is constructed from a triangularization of the
>         finite manifold. Even in this scenario, the proposed spectral
>         loop extraction method scales better asymptotically.
>
>     2.  \[DSW10\] (Ref. \[15\]) has runtime $\mathcal O(N^4)$,
>         where $N = n + n_1 + n_2$ is the size of $\mathrm{SC}_2$ (the
>         detailed runtime is in the paper); in comparison, papers
>         \[DHM19, WCW+17\] in extracting the persistent
>         cycles use the annotation algorithm of \[BCC+12\].
>         The annotation stage has time complexity of
>         $\mathcal O(N^{2.37\cdots})$. As discussed in the main paper,
>         they are tractable for the simplicial complexes built from
>         triangularization; however, this is not the case for VR complex
>         since $n_2$ will grow fast as $n$ increases.
>
>     3.  Lastly, \[FT13, DFW13, CEN09\]
>         have relatively fast runtime (in the order of $n\log n$);
>         however, they either assume that the points are sampled from
>         surfaces in 3D (more precisely, a genus-$g$ surface due to the
>         classification theorem) and/or the graph can be triangularized.
>         It is oftentimes not the case for noisy data sampled from a
>         high-dimensional torus, see, e.g., `ETH` or `MDA` in Figure 3 of
>         the main text. See also the discussion of Ref. \[16\] (bullet
>         point #8) with Reviewer #1 (auuC).
>
> 3.  Thanks for the suggestions for the application through Hodge
>     decomposition; we are aware of these interesting papers, and we will
>     try to include a discussion in our revised paper. As a side note,
>     the applications \[JLYY11, ZWC+20\] through Hodge
>     decomposition uses the information not only from the homology spaces
>     but also the gradient (the image of the down Laplacian
>     ${\mathbf{\mathcal L}}_k^{\rm down}$) and the curl (the image of
>     the up Laplacian ${\mathbf{\mathcal L}}_k^{\rm up}$). Please also
>     refer to our discussion (bullet point #11) with Reviewer 1 (auuC) on
>     the graph signal processing algorithms, in which the information in
>     gradient and curl spaces is also used.
>
> 4.  Indeed, the persistent spectral methods
>     \[WNW20, MX21\] are intriguing works, and we were
>     unaware of these advances. We conjecture that our method can be
>     extended to analyzing a "sequence" of null space embedding
>     ${{\mathbf{Z}}}_\delta$ and shortest loops
>     ${\mathcal C}_i^\delta$'s for different filtration values
>     $\delta$'s. Locating the value of $\delta$ that changes the
>     embedding of the homology basis ${{\mathbf{Z}}}$ or the loops
>     might reveal useful information. One might be able to find some
>     intersections between these two fields! We will for sure incorporate
>     the discussion into our final work.
>
> \[BCC+12\] Oleksiy Busaryev, Sergio Cabello, Chao Chen, Tamal K. Dey, and Yusu Wang. Annotating simplices with a homology basis and its applications. In *Scandinavian Workshop on Algorithm
> Theory*, pages 189–200. Springer, 2012.
>
> \[CEN09\] Erin W. Chambers, Jeff Erickson, and Amir Nayyeri. Mini- mum cuts and shortest homologous cycles. In *Proceedings of the Twenty-Fifth Annual Symposium on Computational Geometry*, pages 377–385, 2009.
>
> \[DFW13\] Tamal K. Dey, Fengtao Fan, and Yusu Wang. An efficient com- putation of handle and tunnel loops via Reeb graphs. *ACM Transactions on Graphics (TOG)*, 32(4):1–10, 2013.
>
> \[DHK11\] Tamal K. Dey, Anil N. Hirani, and Bala Krishnamoorthy. Opti- mal Homologous Cycles, Total Unimodularity, and Linear Programming. *SIAM Journal on Computing*, 40(4):1026–1044, Jan- uary 2011.
>
> \[DHM19\] Tamal K. Dey, Tao Hou, and Sayan Mandal. Persistent 1-cycles:
> Definition, computation, and its application. In *International Workshop on Computational Topology in Image Context*, pages
> 123–136. Springer, 2019.
>
> \[DSW10\] Tamal K. Dey, Tao Hou, and Sayan Mandal. Computing min- imal persistent cycles: Polynomial and hard cases. In *Proceedings of the Fourteenth Annual ACM-SIAM Symposium on Discrete Algorithms*, pages 2587–2606. SIAM, 2020.
>
> \[FT13\] Xin Feng and Yiying Tong. Choking loops on surfaces. *IEEE transactions on visualization and computer graphics*, 19(8):1298–1306, 2013.
>
> \[JLYY11\] Xiaoye Jiang, Lek-Heng Lim, Yuan Yao, and Yinyu Ye. Sta- tistical ranking and combinatorial Hodge theory. *Mathematical Programming*, 127(1):203–244, March 2011.
>
> \[LS14\] Yin Tat Lee and Aaron Sidford. Path finding methods for linear programming: Solving linear programs in o (vrank) iterations and faster algorithms for maximum flow. In *2014 IEEE 55th Annual Symposium on Foundations of Computer Science*, pages 424–433. IEEE, 2014.
>
> \[MX21\] Zhenyu Meng and Kelin Xia. Persistent spectral–based machine learning (PerSpect ML) for protein-ligand binding affinity prediction. *Science Advances*, 7(19):eabc5329, 2021.
>
> \[WCW+17\] Pengxiang Wu, Chao Chen, Yusu Wang, Shaoting Zhang, Changhe Yuan, Zhen Qian, Dimitris Metaxas, and Leon Axel. Optimal topological cycles and their application in cardiac trabeculae restoration. In *International Conference on Information Processing in Medical Imaging*, pages 80–92. Springer, 2017.
>
> \[WNW20\] Rui Wang, Duc Duy Nguyen, and Guo-Wei Wei. Persistent spectral graph. *International journal for numerical methods in biomedical engineering*, 36(9):e3376, 2020.
>
> \[ZWC+20\] Rundong Zhao, Menglun Wang, Jiahui Chen, Yiying Tong, and Guo-Wei Wei. The de Rham–Hodge Analysis and Modeling of Biomolecules. *Bulletin of mathematical biology*, 82(8):1–38, 2020.

---

### Official Review · Reviewer_szyP · 2021-07-08

**Rating:** 7
**Confidence:** 3

**Summary:**

This paper studies $k$-th homology vector spaces through the lens of $k$-Laplacian, a generalization of the usual graph Laplacian to higher order combinatorial structures.
It studies in particular the (fairly general) context of decomposable manifold as a sum of prime manifolds, showing (losely speaking) that under some assumptions (sparse connections), the homology space of a manifold can be obtained as the sum of the homology spaces of the corresponding prime manifolds.
Building on Independent Component Analysis technique, it then provides a way to identify the subspaces of the homology basis computed on the whole manifold to recover the basis of the corresponding prime manifolds.
This has applications in particular in homologous loop detection (detecting the shortest generating element in H1), as showcased on both synthetic and real datasets.


**Limitations And Societal Impact:**

The paper discusses some of its limitations/important requirements (e.g. having access to a Simplicial Complex with the correct homology, etc.). I did not identify any significant potential social impact the paper could have, aside (as mentioned by the paper) from the risk (of any ML algorithms) of being used without proper understanding of it (leading to misinterpretation, etc.).

**Main Review:**


Clarity:
Overall, the paper is well written. It manages to remain concise while preserving mathematical rigor (required due to the theoretical content of the paper), and give enlightening interpretations through the paper.

Significance:
This paper addresses important questions (understanding/representation of higher-order homology vector spaces ; computation of homologous loops...). It provides mathematical analysis, practical algorithms and their implementation (*), and showcase the approach on different datasets in a fairly convincing way.

Questions:
- The paper claims better algorithmic complexity (line 277) for shortest loop computation. Is it reflected in practical running time? I noticed that a running time of ~30sec is mentioned for Alg 2 (line 319), how does the approach developed in [15] performs there?


Minor remarks and suggestions:
- Perhaps including an illustration of the notions introduced in Section 2 (background) would be helpful, if there is enough room.
- (line 64) (typo) I think it should be $B2_{[yz]}(x,y,z) = 1$ the second time.
- (line 85) (typo) $[y_1, ..., y_{beta_1}]$ should be _beta_k. A similar typo occurs in other places.
- (line 119) Talking about the k-Laplacian ``"of M'' is somewhat confusing as those notions/notations are introduced for discrete objects.
- (line 120) (suggestion) Perhaps using \widehat would help readability.
- (line 169) (typo) I think M should be in $\mathbb{R}^D$ (not $\mathbb{R}^{n \times D}$).
- (line 183) (typo) "the the".
- (*) For your information, I tried to run the code. However, I got the following error:
```
  File "..../laplacians.py", line 181, in coboundary_map
    indptr = np.arange(simplices.shape[0]+1, dtype=int) * simplices.shape[1]
IndexError: tuple index out of range
```
after data generation. I may have done something wrong though.

**Time Spent Reviewing:**

10

---

> ### Author Response · Authors · 2021-08-10
> **Authors' Response to Reviewer 3 (szyP)**
>
> Thanks for all the suggestions and comments provided! We will make sure
> to amend all the typos accordingly and clarify the continuous operator
> $\Delta_k$!
>
> 1.  ***"\...how does the approach developed in \[15\] performs
>     there?"*** We intended to compare the result with \[DSW10\]
>     (Ref \[15\]); unfortunately, we contacted the authors several times
>     but cannot get their response (see
>     <https://web.cse.ohio-state.edu/~dey.8/shortloop.html>). We continue
>     trying to contact them and will add the numerical runtime comparison
>     if possible. Nevertheless, we will incorporate more discussions of
>     the related papers on homology loop detection Reviewer #4 (R3gu)
>     provided (bullet point #2).
>
> 2.  ***"I tried to run the code. However, I got the following
>     error\..."*** **Thank you so much** for trying to run the code and
>     pointing this out! I ran the attached rerun the attached python
>     script (in the supplementary material zip file) by "python
>     run_synthetic_manifolds.py", but I cannot reproduce the error. It
>     seems like the error appears when the "simplices" is a 1D array. (It
>     should be a 2D array of size ($n_k$, $k$). We are more than happy to
>     discuss with you during the rolling discussion session to help you
>     solve the bug!
>
> \[DSW10\] Tamal K. Dey, Jian Sun, and Yusu Wang. Approximating loops in a shortest homology basis from point data. In *Proceedings of the Twenty-Sixth Annual Symposium on Computational Geometry*, pages 166–175, 2010.

---

> > ### Comment · Reviewer_szyP · 2021-08-23
> > **Thanks**
> >
> > Thank you for your response.
> >
> > Regarding the code issue, i set up a `conda env` with the required libraries and then simply run `python run_synthetic_manifold.py`. I tried investigating briefly ; the `load_data` step runs but I get:
> > ```python
> > Generating data...
> > 100%
> > Compute L1, Y, Z, and the shortest loops...
> >   0%|                                                                                                                                                                                | 0/1 [00:00<?, ?it/s]
> > n1 = 63824
> > dat.scx.fit() done  # Additional print I added
> > simplices.shape is: (2, 2)  # Additional print
> > simplices.shape[1]: 2  # Additional print
> > number of triangles too small, do not estimate!
> > km1_simplices.shape =  (4390, 1)  # Additional print I added
> > kp1_simplices.shape =  (0,)  # Additional print --> maybe the problem comes from there?
> > k_simplices.shape =  (0,)  # Additional print --> maybe the problem comes from there?
> > dat.hodge_lapla =... done  # Additional print
> > simplices.shape is: (0,)
> > Traceback (most recent call last):
> > ...homology_emb/laplacians.py", line 186, in coboundary_map
> >     print("simplices.shape[1]:", simplices.shape[1])
> > IndexError: tuple index out of range
> > ```
> > So I guess something unexpected happens when calling `dat.hodge_lapla.weighted_Laplacian` in `factorize_emb_and_find_shortest_loop`. But perhaps it is directly at the `load_data` stage, not really clear to me.
> >
> > Anyway, I was just trying the code out of curiosity, I am pretty sure I did something wrong somehow, or that the bug can be solved easily.

---

> > > ### Author Response · Authors · 2021-08-26
> > > **Thanks for the response!**
> > >
> > > We would like to first thank you for your response!
> > >
> > > We created a new environment and installed all the packages; now, we can reproduce the bug. This bug comes from the fact that `dat.scx.triangles` is an empty array (you can verify it by `print(dat.scx.triangles.shape)`, it will show shape = `(0,)`). After some investigations, it seems to appear after the change of `gudhi`'s API from `3.0.0` (the version we used when generating the code) to some other newer versions (e.g., we tested on `3.4.1`). In L49-51 of `homology_emb/complexes.py` with version `3.0.0`, the function `vr_spx_tree.get_skeleton` returns a list; however, in the newer versions, a generator object is returned. Therefore, `edges_alpha` and `triangles_alpha` will both be empty arrays for the latest `gudhi` (since the corresponding generators have already been iterated).
> > >
> > > An easy fix is to explicitly convert the generator objects in L49-51 of the file `homology_emb/complexes.py` to lists by
> > > ```
> > > 49: nodes_ = list(vr_spx_tree.get_skeleton(0))
> > > 50: edges_ = list(vr_spx_tree.get_skeleton(1))
> > > 51: triangles_ = list(vr_spx_tree.get_skeleton(2))
> > > ```
> > > Or you can downgrade `gudhi` to version `3.0.0`. Please let us know if these approaches solve the bug!

---

### Official Review · Reviewer_viBo · 2021-07-15

**Rating:** 7
**Confidence:** 4

**Summary:**

This paper presents a method for calculating (or rather *estimating*)
the null space of the $k$th Laplacian. This enables the analysis of the
geometry of $k$th homology embedding, making it possible to factorise
such an embedding into different subspaces corresponding to individual
prime factors of a given manifold. The proposed method also turns out to
be applicable to improve loop homology generators, leading to geometrically
more concise ('localised') representations of such loops.

The new method is showcases on a variety of data sets, demonstrating its
overall utility for loop localisation and geometric decompositions.


**Limitations And Societal Impact:**

Limitations are partially addressed; a more in-depth discussion of
complexity would be beneficial, in particular since the proposed method
for loop localisation is compared favourably to existing algorithms. One
necessary restriction wrt. the dimension $k$ should also be discussed;
it seems that the decomposition generalises to arbitrary $k$, while the
loop localisation does not. Is this correct?

Being of a primarily theoretical nature, I foresee no adverse societal
effects arising from this work.

**Main Review:**

# Summary of the Review

This paper deals with a highly interesting and intriguing subject,
namely the utility of higher-order Laplacian operators (or Hodge
Laplacians in general) for data analysis. I am excited to see more
papers discussing these concepts in a machine learning context.

That being said, I am a little bit on the fence when it comes to
publishing this paper; the current write-up is very 'dense' and
necessitates knowledge about a lot of additional topics that are not yet
part of the common machine learning canon. Thus, it is possible that the
results of this paper will remain inaccessible to a non-expert audience.
At the same time, I also empathise with efforts of proposing frameworks
that are hitherto unknown; getting the proverbial 'foot in the door' is
no simple endeavour.

I therefore currently lean towards *accepting* this paper, on the
proviso that the paper is rewritten to improve clarity and the logical
flow of the exposition. Please see below for more in-depth comments.

## Detailed Comments on Clarity

- The vector space perspective for $\mathcal{H}_k$ is justified, but
  not necessarily in line with standard topology textbooks, which seem
  to prefer the group perspective, i.e. they are typically referring to
  $\mathcal{H}_k$ as the $k$th homology group. I would suggest to at
  least clarify this distinction briefly.

- In TDA, one typically extracts more than just the ranks of the
  homology groups; please see a recent survey ([Hensel et al., A Survey of Topological Machine Learning Methods](https://www.frontiersin.org/articles/10.3389/frai.2021.681108/full))
  for more details.

- The choice of orientation should not matter for the subsequent
  definition of the Hodge Laplacian, as far as I understand. If so, this
  could be briefly discussed in the text.

- $\mathbf{\omega}_k$ needs a better definition; I found this to be hard
  to understand on my first read of the paper; the definition given in
  l. 53 looks tautological at first glance.

- When introducing the orientations in a boundary matrix, the term
  $\mathbf{B}_2$ in l. 64 appears twice; I think the second term should
  use $y, x$ instead of $x, y$.

- Overall, this discussion could be shortened; I would suggest to either
  provide a brief (graphical) example here instead.

- Is the discussion of continuous operators (in l. 89--) pertinent for
  this manuscript? I appreciate this link but it slightly detracts from
  the overall logical flow of this section. Please consider moving it
  somewhere else (or relegating it to an extended discussion section).

- When discussing the connected sums of manifolds, I would mention that
  this decomposition is unique *up to homeomorphism*.

- When discussing the connected sum representation of manifolds, readers
  with an ML background might appreciate a link to 'disentangled
  representations' here; while 'disentanglement' is a more general
  concept, I think the connection to such a sum will be apparent and
  help build some intuition.

- When discussing definitions, it would sometimes be helpful to explain
  what the assumptions are. For instance, in l. 116, I would start the
  definition with 'Given data $\mathbf{X}$ sampled from a manifold' or
  'We assume that we have...' or something like this.

- In terms of terminology, I am wondering whether the notation with an
  additional hat is really required; does the additional superscript not
  characterise the relevant terms here? In other words, does
  $\mathcal{L}^{(i)}_k$ not suffice to indicate the Laplacian of some
  manifold $\mathcal{M}_i$?

  (feel free to disregard this comment; I merely want to stress that any
  simplifications or updates to the terminology will surely be
  appreciate by readers; there are no *errors* in here, just a lot of
  concepts to keep track of)

- Figure 1 should be shown more prominently; in its present size,
  differences are very hard to grasp.

- What is meant by the 'geometric structure of $\mathcal{H}_k$?
  A priori, these homology groups do not have any geometric structure
  because they are defined in a purely combinatorial manner. All the
  geometrical information requires linking the respective simplicial
  complex to a point cloud etc.

- How strong of an assumption is the condition described in l. 189? The
  subsequent sentences describe this partially, but I was wondering
  about the practical implications here.

- The notation in Assumption 3 cold be explained; I think the main idea
  is to select certain subsets from a matrix, is this correct?

- I fail to understand the discussion on sparsely-connected manifolds.
  Is this related to the sampling density of the manifold? This
  discussion should be improved; I am in particular wondering what the
  meaning of this would be in the point cloud setting.

- Please provide all definitions required to understand Theorem 1 in the
  main text. The main text should be as self-contained as possible.

- Is the bound in Equation (2) tight?

- Please provide more details for the algorithm; I really appreciate the
  pseudo-code here as it serves to demonstrate the practical utility of
  the proposed framework. Nevertheless, maybe the exposition could be
  tipped in favour of explaining the algorithmic details, relegating
  some of the more advanced concepts to a section that addresses readers
  that are more familiar with this background.

- When classifying 2-manifolds, typically, higher-dimensional homology
  groups are employed. The torus is easily distinguished from the circle
  by $\mathcal{H}_2$. The discussion in l. 285 should take this into
  account.

- Proposition 4 could be relegated to the supplements; it does not seem
  to be directly connected to the proposed applications.

- Why are the loops in Figure 2 (k), (l) so localised? Is there
  a preference for a certain basepoint in the algorithm?

- The loop detection seems to be of independent interest and *not* tied
  to a specific decomposition. It might be worth expanding on this.

## Language & Style

The writing in this paper is very goods and flows very well; some of the
formulations are slightly terse, but this can be easily rectified in
a revision. Here are some formulations that I stumbled over upon reading
this paper for the first time:

- 'as a perturbation to the direct sum' --> should this be rephrased as 'as a perturbation of the direct sum'?

- 'theoretical analysis in $\mathbf{Y}$' --> 'the theoretical analysis of $\mathbf{Y}$'

- '$\mathrm{SC_2}$ = [...] commonly used' --> '$\mathrm{SC_2}$, which is commonly used'

- 'choosing an orientation to' --> 'choosing an orientation for'

- 'simplexes' --> 'simplices' (I think both spelling are fine, but
  please ensure consistent use)

- l. 84: 'homology class' --> 'homology classes'

- l. 114: 'how $\mathbf{Y}$ relates to that of each prime manifold':
  what does 'that' refer to here? Is this a reference to the local
  homology embedding of each manifold $\mathcal{M}_i$?

# Update (rebuttal phase)

In light of the detailed responses by the authors, which helped clarify aspects in the paper tremendously, I am happy to raise my score accordingly. I am sure that this publication will make an excellent contribution to NeurIPS.

**Time Spent Reviewing:**

6

---

> ### Author Response · Authors · 2021-08-10
> **Authors' Response to Reviewer 2 (viBo)**
>
> Thanks for the suggestions on the clarification of various
> topics/concepts, i.e., ${\mathcal H}_k$, the choice of orientation, the
> definition of cochains ${\mathbf{\omega}}_k$, the continuous
> operators $\Delta_k$ (L89), the prime decomposition vs. homeomorphism,
> the concept of "disentanglement", and the language & style. We will
> clarify and change accordingly in our final version.
>
> 1.  ***"In TDA, one typically extracts more than just the ranks of the
>     homology groups\..."*** We agree and are aware that not only *rank*
>     information is extracted in the TDA methods. We did not go into
>     details in our original version for the lack of space, but we are
>     sorry for the confusion. We will cite the paper \[HMR21\]
>     and amend the discussion accordingly!
>
> 2.  Thanks for picking that out! In L64 the second
>     $[x,y]$ should be $[y, z]$.
>
> 3.  ***"\...I am wondering whether the notation with an additional hat
>     is really required;"*** The reason why we put a hat on each notation
>     for the disjoint manifold is that we want to distinguish $\Sigma_k$
>     and $\hat\Sigma_k$ (Definition in L122). As discussed with Reviewer
>     1 (auuC) (bullet point #3), this notion is important to rigorously
>     define ${\mathfrak D}_k$ and ${\mathfrak C}_k$, i.e.,
>     ${\mathfrak N}_k = \Sigma_k\cap \hat\Sigma_k$,
>     ${\mathfrak C}_k = \Sigma_k\backslash{\mathfrak N}_k$, and
>     ${\mathfrak D}_k = \hat\Sigma_k\backslash{\mathfrak N}_k$.
>
> 4.  ***"How strong of an assumption is the condition described in l.
>     189?"*** The assumption in L189 holds, for instance, for the case
>     when the intrinsic dimension $d > k$ (L191). This assumption is
>     fairly practical since most of the use cases (e.g., finding loops in
>     a 3D surface) meet this criterion. For example, we can analyze the
>     loop space in a manifold ${\mathcal M}$ with $d \geq 2$; we can also
>     inspect the cavity space ($k=2$) for ${\mathcal M}$ with $d\geq 3$.
>
> 5.  ***"The notation in Assumption 3 could be explained\..."***
>     Intuitively, the conditions say that (for $k=1$) the relative
>     perturbation (induced by connected sum) on the degree of each
>     (non-intersecting) node/edge in ${\mathfrak N}$ is bounded. In our
>     proof of Theorem 1, we "copy" the entries of ${\mathfrak C}_k$ from
>     ${\mathbf{\mathcal L}}_k$ to $\hat{\mathbf{\mathcal L}}_k$
>     (and also ${\mathfrak D}_k$ from $\hat{\mathbf{\mathcal L}}_k$ to
>     ${\mathbf{\mathcal L}}_k$) so that the degree of the simplices in
>     ${\mathfrak N}_k$ will not change too much. Requiring a small
>     perturbation of this subset ${\mathfrak N}_k$ results in a small
>     bound in
>     $\|{\mathbf{\mathcal L}}_k - \hat{\mathbf{\mathcal L}}_k\|_2$.
>     See also our discussion (bullet point #3) with Reviewer 1 (auuC).
>
> 6.  The ***"sparsely connected manifold"*** part is in Assumption 3; it
>     essentially says that $\epsilon_\ell$ should be small (the small
>     perturbation in the non-intersecting simplices ${\mathfrak N}_k$).
>     For this assumption to hold, the density at the connected sum region
>     is small enough compared to other regions (which we called *sparsely
>     connected*).
>
> 7.  ***"Is the bound in Equation (2) tight?"*** The bound is not tight
>     in the sense that we ignore the
>     $\|\mathsf{DiffL}_k^{\rm down} + \mathsf{DiffL}_k^{\rm up} \|_F^2$
>     part (L620 in Supplement). In practice, however, we have
>     $\beta_k\ll n_k$; under this condition, the operator norm
>     $\|\mathsf{DiffL}_k^{\rm down} + \mathsf{DiffL}_k^{\rm up} \|_2^2$
>     will be much smaller than the Frobenius norm
>     $\|\mathsf{DiffL}_k^{\rm down} + \mathsf{DiffL}_k^{\rm up} \|_F^2$,
>     making this bound in (2) almost tight for most use cases. See also
>     the discussion in Ref \[45\].
>
> 8.  ***"When classifying 2-manifolds, typically, higher-dimensional
>     homology groups are employed."*** You are correct. However, using
>     our method can generally save more computational/memory overheads
>     since inspecting the second homology group needs at least the
>     calculation/storage of tetrahedrons from the neighborhood graph.
>     Nonetheless, this is an interesting and important notion to point
>     out; we will make sure to incorporate this discussion into our final
>     version.
>
> 9.  ***"Why are the loops in Figure 2 (k), (l) so localised?"*** We
>     would like to point out that the localization of the loops is the
>     desired goal of our algorithm. In Figure 2l, the algorithm takes the
>     independent basis ${{\mathbf{Z}}}$ (corresponds to the correct
>     prime decomposition) as the input, thus resulting in several
>     "independent" loops; by contrast, in Figure 2k, the extracted loops
>     are coupled together due to the (coordinate-wise) dependency in
>     ${{\mathbf{Y}}}$. The fact that the extracted loops in Figure 2l
>     are similar to each other is due to the way we generate the data;
>     namely, the noise in each torus is not independent. ***"Is there a
>     preference for a certain basepoint in the algorithm?"*** In
>     Algorithm 2, we searched for all edges with large
>     ${{\mathbf{z}}}_i$ values (Line 9 of Algorithm 2). A variant of
>     Algorithm 2 that is based on a certain initial edge (the edge with
>     maximum ${{\mathbf{z}}}_i$) is in Algorithm S1 in Supplement.
>
> 10. ***"The loop detection seems to be of independent interest and not
>     tied to a specific decomposition."*** Algorithm 2 technically is
>     tied to a particular (family of) decomposition; specifically,
>     running Algorithm 2 with the coupled basis ${{\mathbf{Y}}}$ will
>     result in a set of coupled loops (Figure 2k); on the contrary,
>     extracting loops from the independent basis ${{\mathbf{Z}}}$ will
>     yield decoupled loops (Figure 2l).
>
> 11. ***"\...the decomposition generalises to arbitrary $k$, while the
>     loop localisation does not\..."*** Yes, the interpretation is
>     correct! It is because finding the loop is based on Dijkstra.
>
> \[HMR21\] Felix Hensel, Michael Moor, and Bastian Rieck. A Survey of Topological Machine Learning Methods. *Frontiers in Artificial Intelligence*, 4:52, 2021.

---

> > ### Comment · Reviewer_viBo · 2021-08-12
> > **Thanks a lot!**
> >
> > Thanks for your responses to my queries; they are very detailed and I appreciate the effort you made in replying them. I trust that the revision of the paper will be an excellent contribution to the conference, and I am happy to raise my score accordingly.

---

> > > ### Author Response · Authors · 2021-08-13
> > > **Thanks for the response!**
> > >
> > > Thank you very much for your response! If you have any other questions, feel free to continue posting on this thread!

---

### Official Review · Reviewer_auuC · 2021-07-17

**Rating:** 6
**Confidence:** 4

**Summary:**

This paper studies the kernel of the higher-order combinatorial Laplacian, which equivalently amounts to a study of higher-order homology. In particular, given finite samples from a manifold with a ground truth decomposition into a connected sum of smaller ("prime") manifolds, they study the question of when it is possible to computationally recover a "nice" homology basis such that the basis vectors are resolved into independent components contributed from each of the prime manifolds. In this sense, the theoretical aim is quite nice and fundamental. The authors are able to provide a positive answer to their question in the setting where the prime manifolds are connected over a small, sparsely connected region. The proof technique is similar to that for obtaining bounds in spectral graph theory, but is carefully implemented because the problem has lots of block matrices floating around. Finally, the authors propose algorithmic applications of this method to the problem of detecting shortest loops in data, and exemplify their constructions on several point cloud and image datasets.

**Limitations And Societal Impact:**

I believe that the authors have addressed the societal impact quite nicely. I have some concerns about the limitations that I have described above in the "Quality" section.

**Main Review:**

**Originality**: I believe this paper makes an important, foundational step in the study of "spectral topological data analysis" by obtaining recovery guarantees for a nice homology basis as described in the summary. The underlying theoretical problem seems to have been hinted at by reference [17], but I am not aware of works beyond [17] that have taken a close look at this problem. The connection to the geometry of spectral clustering (i.e. the orthogonal cone structure, ref. [36]) provides very nice theoretical motivation.

**Clarity**: Much of the paper is extremely well-written, with careful descriptions as well as adequate explanation and intuition behind assumptions and results. However, I have a few concerns. The mixing of chains/co-chains/cycles/boundaries/coboundaries starts in page 2 and propagates throughout, and thus the reader finds lines that simultaneously talk about cycles and cochains, as well as cochains and boundary matrices (as opposed to coboundary matrices). This gets quite confusing, and occasionally I'm not sure if the paper is talking a particular operator or its adjoint. Page 2 introduces cochains in a boldface definition, but the rest of the paper talks about homology (as opposed to cohomology). The authors put a footnote saying that they use chains/cochains for simplicity, but I feel that mixing terminology between a space and its dual actually makes things complicated. Ref. [17] seems to stick only to chains/boundaries--why not adopt a similar approach, which would save confusion as well as space?

A few more items:
p3, l87: "only identifiable up to a unitary transformation;" please add a reference or explanation

p3 Definitions: I am a little confused: L_k is defined to be the k-Laplacian of M, but later L_k is used as a matrix, which suggests that the underlying space is finite. Should L_k be defined in relation to the data set X? Also, the rest of the definitions here don't refer to the actual data X either

p4,l136: can you expand on the theoretical difficulty? Where would the trivial bound come from?

p4, l168: mismatch between capital D and lowercase d?

p4, Latschev has a theorem stating that if X is a sufficiently fine sample of M, then a particular complex (the Vietoris-Rips complex) will be homotopy equivalent to M. Perhaps this reference can be used to support Assumption 1.

p4, l185: In the definition of $\mathfrak{N}$, i.e. the non-intersecting simplices, I was confused to see that one $\Sigma$ has a hat while the other doesn't. This was clarified in the supplementary materials, but it should be explained in the main text, perhaps in the Section 3 Definitions.

p4, Where are the notions of "destroying" or "creating" simplices defined? Section 2 relegates the definition of connected sum in the case of simplicial complexes to Section 3, but it seems to have been left out from the final version. Also, the notion of "intersecting simplices" is not explained---is this just the union of "created" and "destroyed" simplices, as suggested by p5, l189? Later in the reading process I found more explanations in the supplementary materials, but I would request that the authors be very explicit in setting up these definitions in the main text, as they are crucial for understanding the setup of the main results.

**Quality**: The overall theoretical question and its resolution are both quite nice, and I appreciate the algorithmic aspects that the authors exemplified on multiple datasets. However, some claims may need to be adjusted. In p1, l20: "Topological Data Analysis... aims to extract the dimension of $\mathcal{H}_k$ and has found wide use in analyzing biological..." creates ambiguity about the purpose of the field and the importance of extracting the dimension of $\mathcal{H}_k$. Certainly not all TDA techniques are interested in homology (ref 34 is out of place here, as it purely relies on the Mapper algorithm), and among those that are, perhaps the flagship tool is persistent homology, which goes much beyond simply computing the rank of the homology vector space. In the abstract, the authors write "Our spectral loop detection algorithm scales better than existing methods", and this is true in the context of the complexity of ref. [15] reported in page 6, but what about comparing to the results in ref. [16]? Furthermore, the experiments do not appear to compare runtimes to existing baselines, which I think is a bit of a missed opportunity. I am also curious about the true cost of the eigendecomposition of $\mathcal{L}$. P6L282 suggests that the number of triangles can grow very quickly and pose a challenge to other methods -- doesn't this also affect the computation of $\mathcal{L}_1$, which needs the computation of boundary matrix $B_2$?

**Significance**: Overall, I liked the paper, and I think it has the ingredients of a nice publication. However, I think that the authors could better connect to the ML papers that have come out of graph signal processing research. For instance, reference [A] provided below seems to have some latent connection, in that it utilizes edge flows to cluster a graph and find bottlenecks, which may perhaps be related to the places where prime manifolds connect in the current paper. References [B,C], while less related, relate to useful learning tasks, and connecting to such literature would be helpful for an ML audience.

**Conclusion**: In summary, I like the paper and its theoretical results, but I think it is affected by a few issues of clarity and a somewhat tenuous connection to learning tasks.


[A] Jia, J., Schaub, M. T., Segarra, S., & Benson, A. R. (2019, July). Graph-based semi-supervised & active learning for edge flows. In Proceedings of the 25th ACM SIGKDD International Conference on Knowledge Discovery & Data Mining (pp. 761-771).

[B] Roddenberry, T. M., & Segarra, S. (2019, November). HodgeNet: Graph neural networks for edge data. In 2019 53rd Asilomar Conference on Signals, Systems, and Computers (pp. 220-224). IEEE.

[C] Roddenberry, T. M., Glaze, N., & Segarra, S. (2021, July). Principled simplicial neural networks for trajectory prediction. In International Conference on Machine Learning (pp. 9020-9029). PMLR.

**Time Spent Reviewing:**

7

---

> ### Author Response · Authors · 2021-08-10
> **Authors' Response to Reviewer 1 (auuC)**
>
> We will clarify your suggestions on the concepts of (co)chains, cycles,
> (co)boundaries, and destroying/creating simplices
> (${\mathfrak C}_k$/${\mathfrak D}_k$).
>
> 1.  ***"\...only identifiable up to a unitary transformation:"*** This
>     is true because all harmonic eigenvectors (of
>     ${\mathbf{\mathcal L}}_k$) correspond to the same 0 eigenvalues.
>     To see this, the kernel (null space) of an $n$-by-$n$ matrix
>     ${{\mathbf{A}}}\in\mathbb R^{n\times n}$ is
>     $\ker({{\mathbf{A}}}) = \{{{\mathbf{v}}}\in\mathbb R^{n}: {{\mathbf{A}}}{{\mathbf{v}}}= 0\}$
>     by definition. Let the dimension of the null space be $m$, one can
>     write the null space basis as a projection matrix
>     ${{\mathbf{V}}}= [{{\mathbf{v}}}_1, \cdots, {{\mathbf{v}}}_m]\in\mathbb R^{n\times m}$;
>     here ${{\mathbf{v}}}_i$ is the $i$-th null space basis.
>     Therefore, we have
>     ${{\mathbf{A}}}{{\mathbf{V}}}= {\mathbf{0}}\in\mathbb R^{n\times m}$
>     by definition. Note that
>     ${{\mathbf{U}}}= {{\mathbf{V}}}{{\mathbf{O}}}$ for some
>     unitary matrices ${{\mathbf{O}}}\in\mathbb R^{m\times m}$ will
>     also be a valid basis since
>     ${{\mathbf{A}}}{{\mathbf{U}}}= {{\mathbf{A}}}{{\mathbf{V}}}{{\mathbf{O}}}= {\mathbf{0}}$.
>
> 2.  ***"\...L_k is defined to be the k-Laplacian of M, but later L_k is
>     used as a matrix\..."*** Sorry for the confusion!
>     ${\mathbf{\mathcal L}}_k$ is a *discrete* $n_1\times n_1$ matrix
>     operating on the space of $k$-cochains; in comparison, $\Delta_k$ is
>     the *continuous* operator acting on $k$-forms of the underlying
>     manifold ${\mathcal M}$ (see also L89--L98). We will highlight the
>     differences between these two operators in our final version.
>
> 3.  The trivial bound in L136 can appear when we do not represent
>     $\hat{\mathbf{\mathcal L}}_k$ and ${\mathbf{\mathcal L}}_k$
>     *over the same space of $k$-simplices*---all simplices being
>     created, destroyed, or unaffected by the connected sum operation.
>     Specifically, if we do not "fill" the entries of ${\mathfrak D}_k$
>     in $\hat{\mathbf{\mathcal L}}_k$ in the matrix
>     ${\mathbf{\mathcal L}}_k$ (and also copy the entries of
>     ${\mathfrak C}_k$ in ${\mathbf{\mathcal L}}_k$ into
>     $\hat{\mathbf{\mathcal L}}_k$), the perturbation between these
>     two matrices
>     ${\mathbf{\mathcal L}}_k - \hat{\mathbf{\mathcal L}}_k$ will
>     be dominated by the elements in ${\mathfrak D}_k$ and
>     ${\mathfrak C}_k$. Therefore, one will get a trivial bound of
>     $\|{\mathbf{\mathcal L}}_k - \hat{\mathbf{\mathcal L}}_k\|_2 \approx \|{\mathbf{\mathcal L}}_k\|_2 \leq \lambda_k$.
>     To avert this problem and get the correct bound, we define these two
>     matrices as on Page 3 of the Supplement; we will discuss them more
>     clearly in the final version of the paper. See also our response
>     (bullet point #5) to Reviewer 2 (viBo)
>
> 4.  The lowercase $d$ represents the *intrinsic* dimension of the
>     manifold ${\mathcal M}$, while the uppercase $D$ is the *ambient*
>     dimension of ${\mathcal M}$.
>
> 5.  **Thanks** for providing the theorem by Latschev [Lat01]!!
>     This theorem gives a nice intuition to Assumption 1 and we will
>     incorporate it in our paper.
>
> 6.  ***"\...is this just the union of "created" and "destroyed"
>     simplices, as suggested by p5, l189?"*** Yes, you are correct. Maybe
>     a better way is to first define the non-intersecting simplices as
>     the intersection between $\Sigma_k$ and $\hat\Sigma_k$ (see the
>     definition of this set in L122), i.e.,
>     ${\mathfrak N}_k = \Sigma_k\cap \hat\Sigma_k$. Then we have
>     ${\mathfrak C}_k = \Sigma_k\backslash{\mathfrak N}_k$ and
>     ${\mathfrak D}_k = \hat\Sigma_k\backslash{\mathfrak N}_k$. We are
>     sorry that we left this out in the previous version. We will make
>     sure that these definitions go into our paper.
>
> 7.  ***"\...not all TDA techniques are interested in homology\..."***
>     This is correct and we are aware of the broad scope of TDA methods.
>     It was an unfortunate simplification due to the lack of space, and
>     we are sorry for the confusion. We will amend accordingly!
>
> 8.  ***"\...what about comparing to the results in ref. \[16\]?"***
>     \[DHM20\] (Ref. \[16\]) has proved that in general,
>     obtaining minimum persistent cycle for $k>1$ (e.g., cavities and
>     higher-dimensional holes) is NP-hard. They showed that for the case
>     of $(k+1)$-*pseudomanifolds*, one can reduce the problem to a
>     minimum-cut problem, which can be solved in $\mathcal O(N^2)$ ($N$
>     is the total number of simplices in $\mathrm{SC}$). The
>     polynomial-time algorithm is restricted only to the pseudomanifolds
>     (a generalization of the surface triangularization to higher $k$).
>     These pseudomanifolds have simple structures such that the number of
>     higher-order simplices (e.g., triangles) does not grow too fast as
>     sample size $n$ increases. Unfortunately, this is generally not the
>     case even for noisy data sampled from a high-dimensional torus, see,
>     e.g., `ETH` or `MDA` in Figure 3 of the main text. See also our
>     discussion (bullet point \# 2c) with Reviewer 4 (R3gu). Nonetheless,
>     thanks for pointing this out! We will incorporate them in the final
>     version of the paper.
>
> 9.  ***"\...the experiments do not appear to compare runtimes to
>     existing baselines\..."*** We tried to reach out to the authors of
>     Ref. \[15\] for the codes (in
>     <https://web.cse.ohio-state.edu/~dey.8/shortloop.html>), but they
>     did not respond. We will follow up with them again and add the
>     numerical experiments if possible, since we are also interested to
>     see the outcome of this comparison.
>
> 10. ***"\...doesn't this also affect the computation of
>     ${\mathbf{\mathcal L}}_1$, which needs the computation of
>     boundary matrix ${{\mathbf{B}}}_2$?"*** Indeed, the boundary
>     matrix has time complexity being linear in terms of the number of
>     triangles $\mathcal O(n_2)$. In comparison, \[DSW10\] (Ref.
>     \[15\]) also takes the simplicial complex as an input; building the
>     VR complex needs $\mathcal O(n_2)$ of runtime. Since both methods
>     need this step, we omit this part in our comparison.
>
> 11. Thanks for the suggestions on the papers \[JSSB19, RS19, RGS21\]!
>     These methods (along with the applications using Hodge decomposition
>     pointed out by Reviewer 4 (R3gu)) use not only the harmonic
>     information (the null space of ${\mathbf{\mathcal L}}_k$) but
>     also the gradient/curl fields (which correspond to the images of
>     ${\mathbf{\mathcal L}}_k^{\rm down}$ and
>     ${\mathbf{\mathcal L}}_k^{\rm up}$, respectively). Nonetheless,
>     these are all interesting paper to connect our method with, and we
>     will make sure to discuss and reference to them in our final
>     version!
>
>
> \[DHM20\] Tamal K. Dey, Tao Hou, and Sayan Mandal. Computing min- imal persistent cycles: Polynomial and hard cases. In *Proceedings of the Fourteenth Annual ACM-SIAM Symposium on Discrete Algorithms*, pages 2587–2606. SIAM, 2020.
>
> \[DSW10\] Tamal K. Dey, Jian Sun, and Yusu Wang. Approximating loops in a shortest homology basis from point data. In *Proceedings of the Twenty-Sixth Annual Symposium on Computational Geometry*, pages 166–175, 2010.
>
> \[JSSB19\] Junteng Jia, Michael T. Schaub, Santiago Segarra, and Austin R. Benson. Graph-based Semi-Supervised & Active Learning for Edge Flows. In *KDD*, 2019.
>
> \[Lat01\] Janko Latschev. Vietoris-Rips complexes of metric spaces near a closed Riemannian manifold. *Archiv der Mathematik*, 77(6):522– 528, 2001.
>
> \[RGS21\] T. Mitchell Roddenberry, Nicholas Glaze, and Santiago Segarra. Principled simplicial neural networks for trajectory prediction. In *International Conference on Machine Learning*, pages 9020–9029. PMLR, 2021.
>
> \[RS19\] T. Mitchell Roddenberry and Santiago Segarra. HodgeNet: Graph Neural Networks for Edge Data. In *2019 53rd Asilomar Conference on Signals, Systems, and Computers*, pages 220–224. IEEE, 2019.

---

> > ### Comment · Reviewer_auuC · 2021-08-20
> > **Comments after rebuttal**
> >
> > Dear authors, thank you for the careful responses to my comments as well as those of the other reviewers -- I've read them in detail, and I think that with the suggested improvements, the paper would be a very nice contribution to the ML literature. My request would be to please ensure that the notation and technical definitions are completely airtight and avoid potential confusion as much as possible.
> >
> > Overall, the theoretical aim of this paper was the most interesting aspect to me. To this end, the authors may consider citing the following paper, which seems quite related (released very recently, so no concerns regarding novelty):
> >
> > Garcia Trillos, N., He, P., & Li, C. (2021). Large sample spectral analysis of graph-based multi-manifold clustering. arXiv e-prints, arXiv-2107.
> > https://arxiv.org/pdf/2107.13610.pdf
> >
> > Regarding the detection of homologous loops, it's worth noting this paper (best paper award at SGP 2021):
> > https://www.graphics.rwth-aachen.de/publication/03335/
> >
> > Reviewer viBo suggested a connection to disentangled representations; the following paper (ICLR 21) may be related (but please verify):
> > https://github.com/stanfordmlgroup/disentanglement

---

> > > ### Author Response · Authors · 2021-08-26
> > > **Thanks for the response**
> > >
> > > Thank you for the suggestions on these papers!
> > >
> > > The first paper by Trillos et al. is indeed interesting. The multi-manifold assumption is somewhat related to the "connected sum" assumption (Assumption 1 in the main text). Therefore, it will be informative to discuss the similarities and differences in our final version.
> > >
> > > We will incorporate the second paper (Born et al.) with numerous references suggested by Reviewer R3gu in our final version.
> > >
> > >
> > > The third paper by Zhou et al. proposed a metric for the evaluation of disentanglement. Loosely speaking, the goal of disentanglement is to find an embedding such that each coordinate of the final representation corresponds to some human explainable quantities (such as the rotation of an object in images). In some sense, our framework is similar to disentanglement (assuming that each disjoint manifold $\mathcal M_i$ for $i = 1,\cdots,\kappa$ has some human-explainable meaning) since we are interested in finding a "unitary matrix" such that each subspace corresponds to a single disjoint manifold. The main difference is that we try to find the variation in data based on each *disjoint manifold* instead of each *coordinate* as in the disentanglement community.
> > >
> > > Nonetheless, it might be interesting to see if there are any overlaps between our framework and their paper. Specifically, since their evaluation metric is based on the topological features (the Relative Living Times) computed from persistent homology, we might be able to find some connections between these two. We would like to thank you again for pointing us to this paper, and we are more than happy to point out the possible future directions involving disentanglement in our final version!

---

### Author Response · Authors · 2021-08-10
**Authors' Response**

Thanks to all reviewers for the attentive reading of our paper and for providing detailed as well as constructive feedback!! We will incorporate all your recommendations. Here we can only respond to the most stringent of your questions/concerns. Please refer to the Official Comment under each Official Review for more details!

---

### Decision · Program_Chairs · 2021-09-27

**Decision:**

Accept (Oral)

**Comment:**

This paper investigates the reconstruction of a connected sum of (prime) manifolds from finite samples. It exploits the connection between the null space of the combinatorial Laplacian and the homology of a manifold, in order to recover the homological basis corresponding to the prime manifolds. Theoretical results are proven as to when will the reconstruction be successful. The method is applied to the computation of the shortest homologous cycle problem, and results were shown on various datasets.

All reviewers highly rated the paper. They considered it original, well written, providing strong theoretical results, and delivering inspiring connections between topological data analysis and spectral analysis.